# VIRNE: A COMPREHENSIVE BENCHMARK FOR RL-BASED NETWORK RESOURCE ALLOCATION IN NFV

**Tianfu Wang**[1], **Liwei Deng**[2,*], **Xi Chen**[3], **Junyang Wang**[3], **Huiguo He**[4], **Zhengyu Hu**[1],
**Wei Wu**[3], **Leilei Ding**[3], **Qilin Fan**[5], **Hui Xiong**[1,6,*]
[1]The Hong Kong University of Science and Technology (Guangzhou) [2]Aalborg University
[2]University of Science and Technology of China [4]Sun Yat-sen University
[5]Chongqing University [6]The Hong Kong University of Science and Technology
`tianfuwang@mail.ustc.edu.cn`, `lide@cs.aau.dk` & `xionghui@ust.hk` *

## ABSTRACT

Resource allocation (RA) is critical to efficient service deployment in Network Function Virtualization (NFV), a transformative networking paradigm. This task is termed NFV-RA. Recently, deep Reinforcement Learning (RL)-based methods have been showing promising potential to address this combinatorial complexity of constrained cross-graph mapping. However, RL-driven NFV-RA research lacks a systematic benchmark for comprehensive simulation and rigorous evaluation. This gap hinders in-depth performance analysis and slows algorithm development for emerging networks, resulting in fragmented assessments. In this paper, we introduce Virne, a comprehensive benchmarking framework designed to accelerate the research and application of deep RL for NFV-RA. Virne provides customizable simulations for diverse network scenarios, including cloud, edge, and 5G environments. It features a modular and extensible implementation pipeline that integrates over 30 methods of various types. Virne also establishes a rigorous evaluation protocol that extends beyond online effectiveness to include practical perspectives such as solvability, generalizability, and scalability. Furthermore, we conduct in-depth analysis through extensive experiments to provide valuable insights into performance trade-offs for efficient implementation and offer actionable guidance for future research directions. Overall, with its capabilities of diverse simulations, rich implementations, and thorough evaluation, Virne could serve as a comprehensive benchmark for advancing NFV-RA methods and deep RL applications. The code and resources are available at https://github.com/GeminiLight/Virne.

## 1 INTRODUCTION

Network Function Virtualization (NFV) has emerged as an essential enabler for modern networks, such as cloud data centers, edge computing and 5G, due to its remarkable flexibility and scalability (Yi et al., 2018). By transforming traditional hardware-bound network services into flexible software modules, NFV enables the deployment of Virtual Network Functions (VNFs) on general-purpose servers (Zhuang et al., 2020). A central challenge in NFV is Resource Allocation (NFV-RA) that is essential for effective resource management and service quality (Yang et al., 2021). As illustrated in Figure 1, NFV-RA involves mapping service requests (modeled as virtual networks of interconnected VNFs) onto shared physical infrastructure while satisfying constraints. It is an NP-hard combinatorial optimization problem (Rost & Schmid, 2020), highlighting the need for efficient solutions.

Traditional approaches to NFV-RA, such as exact solvers (Chowdhury et al., 2009; Shahriar et al., 2018) that seek optimal solutions or heuristics (Gong et al., 2014; Fan et al., 2023) dependent on manual design, suffer from neither excessive time consumption nor suboptimal performance. Recently, Reinforcement Learning (RL) has shown promise in solving NFV-RA, which enables the autonomous learning of efficient heuristics by interacting with simulated network environments (Haeri & Trajković, 2017; Yan et al., 2020; Wang et al., 2024c; Wu et al., 2024b). This training paradigm

---

*Corresponding authors.

Table 1: Comparison of NFV-RA benchmarks.

| | Supported Simulation | Implemented Algorithms | RL & Gym Support | Evaluation Perspectives | Last Update |
|---|---|---|---|---|---|
| VNE-Sim (Haeri & Trajković, 2016) | Cloud | 3 heuristics | ✗ | Effectiveness | 2014 |
| ALEVIN (Beck et al., 2014) | Cloud | 5 heuristics | ✗ | Effectiveness | 2016 |
| ALib (Rost et al., 2019) | Cloud | 1 exact | ✗ | Effectiveness | 2019 |
| SFCSim (Xu et al., 2022) | Cloud | 3 heuristics | ✗ | Effectiveness | 2022 |
| Iflye (Tomaszek, 2021) | Cloud | 3 heuristics | ✗ | Effectiveness | 2024 |
| Virne (Ours) | Cloud, Edge, 5G Slicing, etc. | 30+ algorithms (10+ non-RL) | ✓ | Effectiveness & 3 Practicality | 2025 |

operates without requiring high-quality labeled datasets, which are difficult to obtain due to the computational complexity and privacy concerns, making it particularly well-suited for NFV-RA.

However, the advancement of RL for NFV-RA is significantly hampered by the lack of standardized, comprehensive benchmarking. On the one hand, existing benchmarks, as summarized in Table 1, are limited to specific scenarios (e.g., cloud) and a narrow range of non-RL methods. On the other hand, the increasing complexity of modern networks leads to fragmented problem definitions and inconsistent simulations, making fair comparisons and robust evaluations difficult. We summarize the current state of NFV-RA research in Table 4 to highlight these issues. Developing a framework to address these issues poses significant technical challenges, including unifying diverse models, standardizing numerous algorithms, and designing comprehensive evaluations. A unified, accessible framework for reproducible research and standardized evaluation is urgently needed.

In this paper, we introduce **Virne**, a comprehensive benchmarking framework for NFV-RA that offers diverse simulations, unified implementations and thorough evaluations. Firstly, Virne is designed to serve as a unified and readily accessible framework for researchers from both the machine learning and networking communities. It provides a highly customizable simulation environment capable of accurately modeling a wide array of NFV scenarios, from cloud data centers to edge and 5G networks, allowing for the exploration of various resource types, constraints, and service requirements. Secondly, Virne features a modular architecture that simplifies the implementation of various NFV-RA algorithms, which facilitates both efficient utilization and new algorithm development. It includes over 30 NFV-RA algorithms, covering both exact, heuristic and advanced learning-based methods. Thirdly, beyond standard performance metrics, Virne enables in-depth analysis through practical evaluation perspectives such as solution feasibility, generalization across varying network conditions, and scalability with increasing problem size. Finally, through extensive empirical studies conducted within Virne, we offer valuable insights into the effectiveness and characteristics of various algorithms, providing data-driven guidance for future research directions and practical deployments.

Our main contributions, aimed at accelerating data-centric ML research in network optimization, are:

- **Comprehensive Simulations**. Virne is the most comprehensive benchmark for NFV-RA to date, along with gym-style environments supporting highly customizable simulations.

- **Streamlined Implementations**. To facilitate community use, we design a modular and easily expandable implementation pipeline and integrate a wide range of NFV-RA algorithms.

- **Emerging Evaluations**. Beyond online effectiveness alone, Virne enables practical evaluations of solution feasibility, generalization across diverse network conditions, and scalability.

- **Insightful Findings**. Extensive results reveal the impact of different modules and nuanced performance comparisons, providing actionable insights for future applied ML research.

## 2 NFV-RA PROBLEM DEFINITIONS

Due to distinct considerations of network scenarios, existing studies formulate NFV-RA varying in system models, constraints, and objectives. To enhance clarity and consistency, we provide a unified definition that combines a basic cost optimization model with extensions for emerging scenarios.

## 2.1 BASIC DEFINITION FOR COST OPTIMIZATION

**System Model**. In NFV, user network services and the physical infrastructure are virtualized as virtual networks (VNs) and physical network (PN), respectively. As illustrated in Figure 1, in online network systems, service requests represented as VNs are continuously arriving at PN, seeking the physical resource with specific service requirements. Each arrived VN and the corresponding snapshot of PN consist of an instance $I = (\mathcal{G}_v, \mathcal{G}_p)$, where the PN $\mathcal{G}_p$ and VN $\mathcal{G}_v$ are modeled as undirected graphs, $\mathcal{G}_p = (\mathcal{N}_p, \mathcal{L}_p)$ and $\mathcal{G}_v = (\mathcal{N}_v, \mathcal{L}_v, \omega, \varpi)$, respectively. Here, $\mathcal{N}_p$ and $\mathcal{L}_p$ denote the sets of physical nodes and links, indicating servers and their interconnections; $\mathcal{N}_v$ and $\mathcal{L}_v$ denote the sets of virtual nodes and links, representing services and their relationships; $\omega$ and $\varpi$ denote the arrival time and lifetime of VN. We denote $C(n_p)$ as available computing resources for physical node $n_p \in \mathcal{N}_p$, and $B(l_p)$ as available bandwidth resources of physical link $l_p \in \mathcal{L}_p$. Besides, $C(n_v)$ and $B(l_v)$ denote the demands for computing resource by a virtual node $n_v \in \mathcal{N}_v$ and bandwidth resource by a virtual link $l_v \in \mathcal{L}_v$. Over a specific period, we collect all instances into a set $\mathcal{I}$.

**Embedding Constraints**. Mapping a VN $\mathcal{G}_v$ onto the sub-PN $\mathcal{G}_{p'}$ is formulated as a mapping function $f_\mathcal{G} : \mathcal{G}_v \to \mathcal{G}_{p'}$. It consists of two sub-mapping processes, i.e., node mapping $f_\mathcal{N}$ and link mapping $f_\mathcal{L}$. Node mapping $f_\mathcal{N}$, aims to find a suitable physical node $n_p = f_\mathcal{N}(n_v)$ to place each virtual node $n_v$ while adhering to the one-to-one mapping and resource availability constraints. Concretely, virtual nodes in the same VN must be placed in different physical nodes and each physical node can host at most one virtual node. And the physical node $n_p$ must have enough available resources to place the corresponding virtual node $n_v$, i.e., $C(n_v) \leq C(n_p), \forall n_v \in \mathcal{N}_v$. Link mapping aims to find a connected

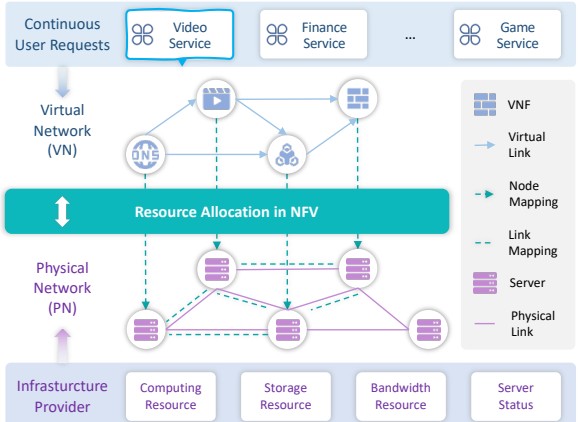

Figure 1: A brief illustration of the NFV-RA problem.

physical path $\rho_p = f_\mathcal{L}(l_v)$ to route each virtual link $l_v$ while satisfying the path connectivity and resource capacity constraints. Concretely, the physical path $\rho_p$ should connect physical nodes hosting the two endpoints of virtual link $l_v$. And each physical link $l_p \in \rho_p$ of physical path $\rho_p$ should have enough bandwidth to route the corresponding virtual link $l_v$, i.e., $B(l_v) \leq B(l_p), \forall l_v \in \mathcal{L}_v, l_p \in \rho_p$.

**Optimization Objective**. Considering the randomness of online service requests, NFV-RA mainly aims to maximize the resource utilization of each instance $I$, which facilitates long-term resource profit and request acceptance (He et al., 2023; Zhang et al., 2023). To assess the solution quality $S = f_\mathcal{G}(I)$ of instance $I$, the revenue-to-cost ratio (R2C) is a widely used metric, defined as follows:

$$\max \text{R2C}(S) = (\varkappa \cdot \text{REV}(S)) / \text{COST}(S). \tag{1}$$

Here, $\varkappa$ is a binary variable indicating the feasibility of the solution $S$: $\varkappa = 0$ if $S$ violates some constraints, and $\varkappa = 1$ otherwise. REV($S$) and COST($S$) denote the generated revenue and incurred cost by embedding VN $\mathcal{G}_v$. If $\varkappa = 1$, REV($S$) denotes the revenue from the VN, calculated as $\sum_{n_v \in \mathcal{N}_v} C(n_v) + \sum_{l_v \in \mathcal{L}_v} B(l_v)$ and COST($S$) denotes the resource consumption of PN, calculated as $\sum_{n_v \in \mathcal{N}_v} C(n_v) + \sum_{l_v \in \mathcal{L}_v} (|f_\mathcal{L}(l_v)| \times B(l_v))$. Here, $|f_\mathcal{L}(l_v)|$ quantifies the length of the physical path $\rho_p$ routing the virtual link $l_v$. See Appendix A.1 for the detailed formulation.

## 2.2 EXTENSIONS IN EMERGING NETWORKS

The application of NFV is pivotal in emerging network scenarios, such as heterogeneous resourcing networks, latency-aware edge networks, and energy-efficient green networks. These scenarios require additional consideration of their unique challenges. To better align NFV-RA with practical requirements, we discuss several extended definitions of NFV-RA in popular scenarios in Appendix A.2.

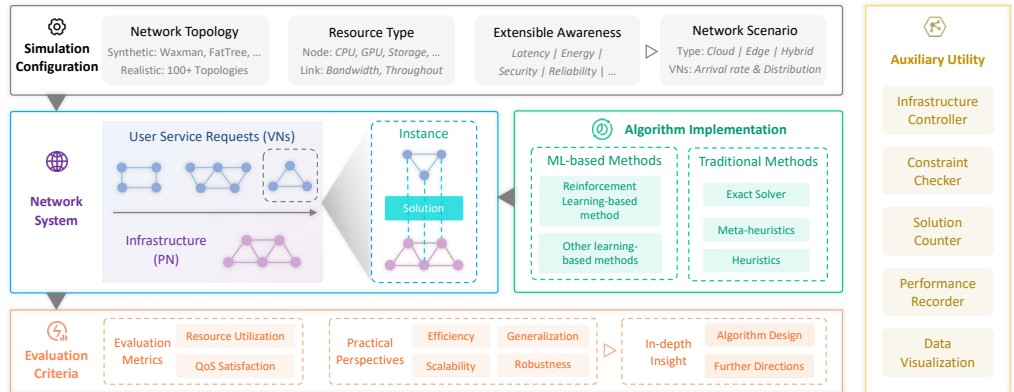

Figure 2: The architecture of Virne benchmark. Virne offers a streamlined workflow for supporting comprehensive experimentation of NFV-RA algorithms. The process begins with customizing simulation configurations that define network scenarios and conditions. Then, the network system is instantiated, triggering a series of service request events to process. At each event, the selected NFV-RA algorithm interacts with the system to resolve the instance. For each simulation, both the processing and final results are automatically recorded for subsequent analysis.

## 3  VIRNE: A COMPREHENSIVE NFV-RA BENCHMARK

To provide high-quality simulation and implementation, we introduce three design principles for Virne, following established software engineering practices (Van Vliet et al., 2008): (a) *Versatile customization* allows the platform to meet diverse simulation needs on varying network scenarios and conditions. (b) *Scalable modularity* built platforms with a modular architecture to support flexible configurations and easy extensibility. (c) *Intuitive usability* prioritizes a user-friendly interface, enabling users to focus on experiment outcomes rather than implementation complexities. Guided by these principles, as illustrated in Figure 2, we implement Virne with five key modules as follows.

### 3.1  SIMULATION CONFIGURATION

Modern network systems are diverse and intricate, associated with various realistic factors such as resource availability and service requirements. To support high customization for simulating different network scenarios and conditions, Virne abstracts both the PN and VN as graphs, with customizable attributes for nodes, links, and the overall network. Specifically, Virne provides the following key customizable elements: (a) *Network Topologies*. Users can select from various topological synthesis methods or real-world physical infrastructure topologies, such as those from SDNLib. (b) *Resource Availability*. Virne enables users to define multiple resource types (e.g., CPU, GPU, bandwidth) and their availability across different levels of the network (i.e., nodes, links and graph), allowing the reflection of the specific resource characteristics of different network environments (c) *Service Requirements*. Users can specify additional service requirements, such as latency, energy efficiency, and reliability, to ensure that the simulation reflects the needs of different network applications. By adjusting these parameters, Virne provides the flexibility to simulate a wide array of network scenarios, including cloud-based infrastructures, edge computing and 5G networks. This customization enables comprehensive testing, accommodating fluctuating resource demands and evolving network conditions. We provide configuration examples in Figure 6 and Appendix C.1.

### 3.2  NETWORK SYSTEM

With the above-mentioned configurations, Virne automatically creates a network system that functions as an event-driven simulator, which mainly consists of a PN as infrastructure and a series of sequential arrived VN requests. Virne treats each VN request arrival as a discrete event that triggers the corresponding resource scheduling and allocation procedures. An event represents the interaction between the VN request $\mathcal{G}_v$ and a snapshot of the PN $\mathcal{G}_p$ at the time of the request's arrival, denoted as an instance $I$. The specific NFV-RA algorithm then solves the corresponding NFV-RA problem for that instance to obtain a solution, $S = f_{\mathcal{G}}(I)$. Next, the system evaluates the feasibility of the solution based on the resource availability constraints and service requirements satisfaction. If feasible, the VN request is accepted; otherwise, it is rejected. This event-driven mechanism is

Figure 3: A unified pipeline of gym-style environment and RL-based NFV-RA methods in Virne.

designed to simulate realistic, complex network environments while enabling efficient handling of diverse network scenarios and fluctuating conditions.

## 3.3 ALGORITHM IMPLEMENTATION

To facilitate direct utilization and further extensions, Virne integrates diverse NFV-RA algorithms, covering both learning-based and traditional methods (see Appendix C.3). These algorithms are systematically organized to streamline the implementation process. Here, we highlight our unified pipeline of gym-style environments and RL-based NFV-RA methods, as illustrated in Figure 3.

### 3.3.1 NFV-RA AS A MARKOV DECISION PROCESS

To address the randomness of online networks, most existing works model NFV-RA solution construction as a Markov Decision Process (MDP), which sequentially selects a physical node to place a virtual node at each decision step $t$, until all virtual nodes are placed or any constraints are violated. Decision sequence defaults to VN node index, with support for customizable node ranking in Virne.

Formally, we define this process as a tuple $(\mathcal{S}, \mathcal{A}, P, R, \lambda)$. Concretely, $\mathcal{S}$ is the state space, where $s_t \in \mathcal{S}$ represents the embedding status of VN and PN. $\mathcal{A}$ is the action space, where $a_t$ referring to a set of physical nodes. $P : \mathcal{S} \times \mathcal{A} \times \mathcal{S} \to [0, 1]$ is the state transition function, indicating the conditional transition probabilities between states. During one transition, the system will attempt to place the selected action, i.e., a physical node $a_t = n_p$ to route the to-be-decided virtual node $n_v$. If node placement succeeds, link routing will be conducted to route the virtual links connecting $n_v$ and its already-placed neighbor nodes. The shortest-path algorithms are used to identify the shortest and constraint-satisfied physical paths for these virtual links. Only when both node placement and link routing are successful, the physical network's available resources are updated with the VNR's requirements. Otherwise, the VN is rejected. $R : \mathcal{S} \times \mathcal{A} \to \mathbb{R}$ is a designed reward function to guided optimization. $\lambda \in [0, 1)$ is the discount factor.

At each decision timestep $t$, observing the state $s_t$ of the environment, the agent takes an action $a_t \sim \pi(\cdot|s_t)$ according to a policy $\pi_\theta$ parameterized by $\theta$. Then, the environment will transit to a new state $s_{t+1} \sim P(\cdot|s_t, a_t)$ and feedback a reward $R(s_t, a_t)$ and. During interactions, a trajectory memory collects these step information as experiences. Using these experiences, the objective of the agent is to learn an optimal policy for resource allocation that maximizes the expected sum of discounted reward, i.e., the solution quality of NFV- $\pi_\theta^* = \arg\max_{\pi_\theta} \mathbb{E}_{\pi_\theta} \left[ \sum_{i=0}^{T} \gamma^i R(s_t, a_t) \right]$

Note that the above MDP introduced is the most adopted version. As a holistic framework, Virne also supports the emerging MDP version in recent studies. Due to the presentation clarity and page limit, we offer other MDP versions of NFV-RA in Appendix C.2, such as Constrained MDP (Wang et al., 2025b) for constraint handling and Multi-task MDP (Wang et al., 2024c) for generalization.

### 3.3.2 UNIFIED PIPELINE FOR EFFICIENT IMPLEMENTATIONS

Considering this widely-used MDP model, we generally unify existing NFV-RA RL-based algorithms into three key components: MDP modeling, policy architecture, and training methods. These methods model the NFV-RA solution construction process as above-introduced MDPs with a specific reward design and feature engineering. Subsequently, they use various neural networks to build policy architectures, such as Convolutional Neural Networks (CNN) (Krizhevsky et al., 2012) and Graph Convolutional Network (GCN) (Kipf & Welling, 2017). These policies are trained with a selected RL method, such as Asynchronous Advantage Actor-critic (A3C) (Mnih et al., 2016) and Proximal Policy Optimization (PPO) (Schulman et al., 2017).

Following this insight, Virne implements NFV-RA algorithms through a modular and extensible pipeline comprising several core components as illustrated in Figure 3. Specifically, an *instance-level environment* enable interactions with the agent for sequentially constructing solutions, where a customized *reward function* provides feedback at each step. The RL agent consists of a *feature constructor*, which extracts relevant features as inputs for the neural networks, and a *neural policy*, implemented with various architectures. To support RL training, an *experience memory* module continuously collects interaction data, while a designated *training method* optimizes the policy. This modular design is central to Virne's ability to integrate diverse approaches seamlessly, due to the high customization of each module. By standardizing the implementation process, Virne ensures consistency, reusability, and reduced complexity of implementation, which accelerates the development of new algorithms by providing a unified pipeline.

### 3.3.3 Implemented RL-based NFV-RA Algorithms

As summarized in Table 4 in Appendix B.2, various studies on NFV-RA often share similar core methodologies in their core RL framework design but differ primarily in terms of the specific network scenarios considered or specialized implementation techniques employed. Virne offers a comprehensive suite of state-of-the-art reinforcement learning methods for NFV resource allocation. The framework flexibly combines various RL training approaches (including MCTS, PG, A3C, and PPO) with different neural network architectures (such as MLP, CNN, S2S, GCN, GAT, BiGCN, BiGAT, and HeteroGAT). It also incorporates various implementation enhancements like custom reward functions, feature engineering combinations, and action masking mechanisms. See Appendix C.3.1 for descriptions of these RL methods, neutral policies, and implementation techniques.

To provide clarity in both implementation and evaluation, we categorize these works under common names that reflect RL algorithms and their neural policy architectures. For example, the PPO-GCN method uses graph convolutional networks as feature encoders while employing Proximal Policy Optimization for efficient training. In the subsequent experiments, we will first investigate the impact of distinct implementation techniques, such as reward function and feature engineering, and identify the most efficient configurations for general implementation. Then, we evaluate their effectiveness in the cloud and other popular scenarios.

### 3.4 Auxiliary Utility

To enhance usability and streamline analysis, Virne includes several key auxiliary utilities, as shown in Figure 2. These are designed to simplify the process of managing simulations and interpreting results, particularly including: (a) A *system controller* manages the simulation of physical and virtual networks, providing method choices for customization; (b) A *solution monitor* tracks solution feasibility and performance during execution, helping users assess whether solutions meet their defined criteria; (c) *Visualization tools* offer interactive and visual representations of simulation results, providing users with an intuitive way to analyze network system behaviors. These utilities support more advanced customizations for algorithm design and result analysis.

### 3.5 Evaluation Criteria

To offer a systematic evaluation, Virne provides a suite of metrics and multiple practical perspectives.

**Performance Metrics**. Virne includes the critical performance metrics of NFV-RA algorithms (Fischer et al., 2013; Wu et al., 2024a), including request acceptance rate (RAC), long-term revenue-to-cost (LRC), long-term average revenue (LAR) and average solving time (AST). See Appendix C.5 for their definitions.

**Practicality Perspectives**. To comprehensively assess the practicality of NFV-RA algorithms, we develop multiple emerging evaluation protocols that extend beyond mere online effectiveness. In particular, we consider three aspects of NFV-RA algorithms: (a) *Solvability* denotes the ability to find feasible solutions; (b) *Generalization* indicates reliable performance across various network conditions; (c) *Scalability* measures how effectively it accommodates increases in network size and complexity. Gaining these insights with our pre-provided evaluation interfaces, Virne helps users understand the practical viability of an algorithm and guides further development and application.

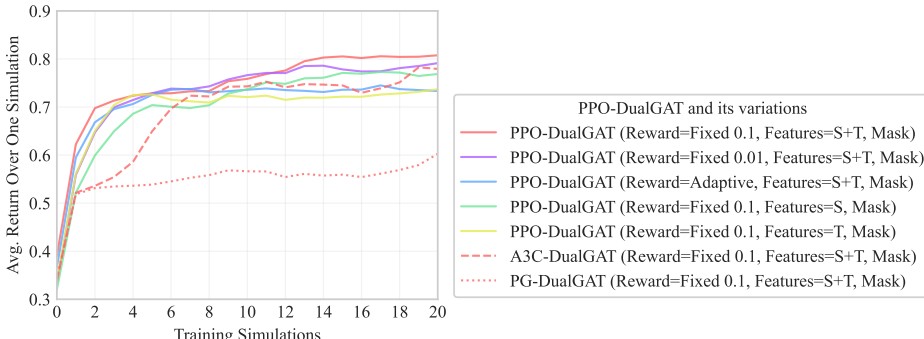

Figure 4: Training curves of PPO-DualGAT and its variations. Each point represents the R2C ratio over all VN Requests within a single training simulation round.

## 4 EMPIRICAL ANALYSIS

### 4.1 EXPERIMENTAL SETUP

We consider the most general scenarios as main network systems, i.e., cloud computing, and describe default settings for simulation and implementation. Subsequently, some parameters may be changed to simulate diverse network scenarios and conditions.

**Simulation Settings**. We adopt the widely-used topologies as physical networks: synthetic WX100 (Waxman, 1988), and real-world GEANT and BRAIN (Orlowski et al., 2010), which cover various network scales and densities. See Appendix D.1.2 for their detailed descriptions. Computing resources of physical nodes and bandwidth resources of physical links are uniformly distributed within the same range of units, i.e., $\mathcal{X}_{C(n_p)} \sim \mathcal{U}(50, 100)$ and $\mathcal{X}_{B(l_p)} \sim \mathcal{U}(50, 100)$. In default settings, for each simulation run, we create 1000 VN with varying sizes following a uniform distribution, $\mathcal{X}_{|\mathcal{G}_v|} \sim \mathcal{U}(2, 10)$. The computing resource demands of the nodes and the bandwidth requirements of the links within each VN are uniformly distributed, i.e., $\mathcal{X}_{C(n_v)} \sim \mathcal{U}(0, 20)$ and $\mathcal{X}_{B(l_v)} \sim \mathcal{U}(0, 50)$, respectively. The virtual nodes in each VN are randomly interconnected with a probability of 50%. The lifetime of each VN follows an exponential distribution with an average of 500 time units. The arrival of these VNs follows a Poisson process with an average rate $\eta$, where $\eta$ denotes the average arrived VN count per unit of time. Due to the varying physical resource supply in these topologies caused by distinct scale and density, we use different $\eta$, i.e., (0.16, 0.016, 0.004) for (WX100, GEANT, BRAIN), to accommodate the reasonable request demand. In the subsequent experiments, we manipulate the distribution settings of VNs and change the PN topologies to simulate various network systems.

**Algorithm Implementation**. In Appendix D.1.1, we provide the details of parameter setting, experimental methods on training and testing, and descriptions of computing resources.

### 4.2 PERFORMANCE COMPARISON

#### 4.2.1 EXPLORATION ON IMPLEMENTATION TECHNIQUES

Due to its training efficiency and strong empirical performance (Wang et al., 2022), we adopt PPO as the default RL algorithm in subsequent experiments. This choice is further supported by the study of *Impact on RL Training Methods* as follows. Furthermore, there are several core implementation choices, such as reward design, feature engineering, and action masking, that substantially affect the performance of RL-based NFV-RA algorithms. To quantify these effects, we systematically evaluate three representative policy architectures (PPO-MLP, PPO-ATT, PPO-DualGAT) on WX100, with results summarized in Table 2. Next, we elaborate on the analysis of each study.

**Impact on RL Training Methods**. We evaluate the impact of different RL training algorithms by comparing the performance of PG, A3C, and PPO. All three methods are paired with the same PPO-DualGAT policy architecture. As illustrated by the learning curves in Figure 4, PPO demonstrates significantly superior performance. It converges rapidly to the highest average return, stabilizing at a high level of solution quality early in the training process. In contrast, A3C shows slower

Table 2: Impact of key implementation techniques. Reward function is specified by its type (fixed or adaptive) and the intermediate reward value (0, 0.01, 0.1, 0.2; where 0 indicates no intermediate reward). Features indicate whether node Status (S) and/or Topological (T) metrics were used (✓ for used, ✗ for not used). Action Mask indicates whether action masking was applied (✓) or not (✗).

| Reward Function | | Features | | Action Mask | PPO-MLP | | | PPO-ATT | | | PPO-DualGAT | | |
|---|---|---|---|---|---|---|---|---|---|---|---|---|---|
| Type | Value | S | T | | RAC | LRC | LAR | RAC | LRC | LAR | RAC | LRC | LAR |
| Fixed | 0.1 | ✓ | ✓ | ✗ | 0.666 | 0.670 | 11745.0 | 0.667 | 0.644 | 12174.0 | 0.648 | 0.750 | 12430.1 |
| Fixed | 0.1 | ✓ | ✗ | ✓ | 0.678 | **0.675** | 11159.0 | 0.660 | 0.667 | 11134.3 | 0.766 | **0.754** | 15054.7 |
| Fixed | 0.1 | ✗ | ✓ | ✓ | 0.703 | 0.654 | 12693.8 | 0.579 | 0.575 | 9323.5 | 0.733 | 0.685 | 13333.1 |
| Adaptive | - | ✓ | ✓ | ✓ | 0.705 | 0.619 | 12139.6 | 0.702 | 0.643 | 12368.6 | 0.772 | 0.744 | 14216.6 |
| Fixed | 0.2 | ✓ | ✓ | ✓ | 0.616 | 0.643 | 11869.4 | 0.706 | **0.675** | 12420.7 | 0.769 | 0.731 | 15045.5 |
| Fixed | 0.01 | ✓ | ✓ | ✓ | 0.709 | 0.645 | 12719.1 | 0.517 | 0.601 | 8339.7 | 0.766 | 0.748 | 14516.6 |
| Fixed | 0 | ✓ | ✓ | ✓ | 0.560 | 0.596 | 8259.5 | **0.716** | 0.658 | 12629.0 | 0.741 | 0.753 | 14047.6 |
| Fixed | 0.1 | ✓ | ✓ | ✓ | **0.719** | 0.647 | **12944.4** | 0.712 | 0.661 | **12657.7** | **0.781** | 0.738 | **15138.6** |

convergence and reaches a lower performance plateau. PG struggles the most, exhibiting the slowest learning rate and the lowest final performance among the three. These findings highlight PPO's sample efficiency and robust performance, justifying its selection as the default RL method for our subsequent experiments.

**Impact on Reward Function Design**. The design and magnitude of intermediate rewards strongly affect training and final performance. Across all solvers, a moderate fixed intermediate reward (`fixed, 0.1`) consistently yields the best or near-best results. In contrast, very small or no intermediate rewards result in clear performance drops, highlighting the importance of sufficient reward signals for effective exploration. Interestingly, the adaptive intermediate reward, which intuitively normalizes the total reward for different VN sizes, does not achieve optimal performance in practice. As illustrated in Figure 4, we observe that a well-tuned fixed reward of 0.1 has higher convergence than the adaptive reward, which validates our initial performance observations. The adaptive reward is also outperformed by a smaller fixed reward of 0.01, which further demonstrates that it may introduce a noisy signal for the optimization process, leading to sub-optimality. This suggests that, ❶ *while normalization is conceptually appealing, a fixed and moderate reward signal is more effective for guiding policy learning in the NFV-RA context.*

**Impact on Feature Engineering Combination**. We compare the use of Status (S), Topological (T), and their combination in the feature constructor. The results show that combining both Status and Topological features (✓, ✓) generally leads to the best or second-best outcomes for acceptance rate and revenue, regardless of the solver architecture. For example, PPO-ATT+ achieves the best acceptance rate (0.712) and LRC (0.661) with both features, and a marked drop when using only one. As illustrated in Figure 4, leveraging these advanced features also contributes to faster and higher convergence. This demonstrates that ❷ *topological metrics serve as a valuable augmentation, capturing global network context and node importance, even for graph neural networks.*

**Impact on Action Mask Strategy**. Applying action masking (✓), which prevents the agent from selecting infeasible actions (e.g., resource availability satisfaction), consistently improves performance across all methods. For instance, in PPO-MLP+ and PPO-DualGAT+, enabling action masking increases acceptance rate by up to 0.053 compared to otherwise identical configurations without masking. This demonstrates that ❸ *explicit constraint enforcement is vital for robust RL-based NFV-RA solutions, since the constraints of NFV-RA are intricate and hard.*

This study reveals that the most effective implementation techniques of the RL-based NFV-RA method are achieved by combining moderate intermediate rewards, comprehensive feature engineering, and action masking. In subsequent experiments, we consider them as the default implementation.

### 4.2.2 EFFECTIVENESS IN ONLINE ENVIRONMENTS

To evaluate the effectiveness of these NFV-RA algorithms, we conduct online simulations across three distinct network topologies: WX100, GEANT, and BRAIN, each with different traffic rates ($\eta$) as specified. Table 3 reports experimental results of the implemented RL-based and traditional algorithms on key metrics. While traditional heuristics are exceptionally fast, they fall short in

Table 3: Performance comparison of implemented RL-based and traditional NFV-RA algorithms.

| Solver | WX100 ($\eta = 0.14$) | | | | GEANT ($\eta = 0.016$) | | | | BRAIN ($\eta = 0.004$) | | | |
|---|---|---|---|---|---|---|---|---|---|---|---|---|
| | RAC↑ | LRC↑ | LAR↑ | AST↓ | RAC↑ | LRC↑ | LAR↑ | AST↓ | RAC↑ | LRC↑ | LAR↑ | AST↓ |
| PPO-MLP | 71.90 | 0.65 | 12944.40 | 0.13 | 55.80 | 0.67 | 645.04 | 0.03 | 51.30 | 0.69 | 155.10 | 0.14 |
| PPO-CNN | 71.70 | 0.65 | 12964.87 | 0.13 | 54.80 | 0.65 | 643.83 | 0.09 | 51.10 | 0.69 | 151.51 | 0.13 |
| PPO-ATT | 71.20 | 0.66 | 12657.69 | 0.14 | 54.50 | 0.65 | 707.01 | 0.10 | 51.00 | 0.68 | 156.40 | 0.15 |
| PPO-GCN | 66.80 | 0.64 | 11462.65 | 0.14 | 58.70 | 0.72 | 763.68 | 0.12 | 49.50 | 0.71 | 125.63 | 0.09 |
| PPO-GAT | 71.90 | 0.70 | 13178.13 | 0.15 | 58.40 | 0.70 | 724.31 | 0.07 | 44.60 | 0.51 | 95.32 | 0.09 |
| PPO-GCN&S2S | 65.80 | 0.63 | 11501.94 | 0.13 | 58.50 | 0.72 | 718.76 | 0.06 | 44.40 | 0.59 | 99.83 | 0.18 |
| PPO-GAT&S2S | 67.90 | 0.67 | 12445.03 | 0.16 | 57.30 | 0.69 | 754.61 | 0.15 | 51.80 | 0.68 | 136.67 | 0.19 |
| PPO-DualGCN | 70.20 | 0.71 | 13467.57 | 0.17 | **60.40** | **0.75** | **791.75** | 0.22 | 54.80 | **0.78** | 176.15 | 0.23 |
| PPO-DualGAT | **78.10** | **0.74** | **14938.60** | 0.18 | 59.10 | 0.72 | 739.27 | 0.10 | **58.90** | 0.75 | **180.78** | 0.13 |
| PPO-HeteroGAT | 72.50 | 0.66 | 12691.03 | 0.27 | 53.30 | 0.66 | 621.47 | 0.30 | 49.30 | 0.66 | 133.52 | 0.38 |
| MCTS | 74.30 | 0.44 | 12642.27 | 3.38 | 48.20 | 0.45 | 494.64 | 2.96 | 40.80 | 0.47 | 83.91 | 3.59 |
| SA-Meta | 65.50 | 0.63 | 10467.60 | 1.58 | 38.60 | 0.62 | 396.49 | 0.49 | 36.10 | 0.58 | 75.50 | 1.13 |
| GA-Meta | 71.70 | 0.59 | 11977.41 | 3.22 | 49.90 | 0.58 | 517.63 | 2.34 | 42.50 | 0.57 | 85.45 | 3.75 |
| PSO-Meta | 69.10 | 0.52 | 10706.48 | 4.29 | 45.30 | 0.46 | 457.93 | 3.68 | 41.50 | 0.46 | 80.67 | 4.20 |
| TS-Meta | 65.70 | 0.66 | 11141.91 | 1.35 | 40.90 | 0.69 | 402.04 | 0.62 | 37.10 | 0.64 | 68.41 | 1.11 |
| NRM-Rank | 60.70 | 0.52 | 9826.94 | 0.07 | 37.90 | 0.51 | 394.29 | 0.03 | 48.30 | 0.64 | 142.99 | 0.04 |
| RW-Rank | 60.10 | 0.56 | 9396.32 | **0.04** | 38.70 | 0.52 | 418.14 | **0.01** | 50.20 | 0.65 | 147.64 | 0.05 |
| GRC-Rank | 58.90 | 0.56 | 9269.03 | **0.04** | 36.70 | 0.47 | 353.21 | **0.01** | 48.40 | 0.64 | 144.55 | **0.03** |
| PL-Rank | 68.10 | 0.67 | 11570.27 | 0.32 | 55.30 | 0.66 | 661.93 | 0.04 | 48.70 | 0.70 | 136.79 | 0.36 |
| NEA-Rank | 64.00 | 0.66 | 10837.51 | 0.83 | 47.20 | 0.62 | 543.18 | 0.11 | 53.90 | 0.70 | 148.70 | 1.46 |
| RW-BFS | 40.00 | 0.57 | 7771.38 | 0.05 | 56.70 | 0.64 | 736.28 | **0.01** | 48.00 | 0.64 | 140.38 | 0.16 |

solution quality. Similarly, meta-heuristics, despite achieving competitive acceptance rates, are hindered by significant computational overhead, making them impractical for real-time environments. This highlights that ❹ *RL-based methods often offer a favorable balance between solution quality and efficiency, making them a compelling choice for NFV-RA*. Advanced RL agents, specifically those with dual graph neural network architectures like PPO-DualGAT and PPO-DualGCN, consistently deliver superior performance. This suggests that ❺ *dual-GNN models' ability to concurrently process and relate features from both the virtual and physical network graphs provides a significant edge in finding high-quality embeddings.* Furthermore, it is notable that ❻ *the topological characteristics of PN, such as scale and density, impact the relative performance of NFV-RA algorithms, particularly GNN architectures.* For instance, on the smaller-scale GEANT and BRAIN, GCN-based methods achieve more competitive and even better performance. This reveals that GATs with an adaptive aggregation mechanism may be more advantageous in denser topologies for prioritizing critical connections, but provide less improvement in sparse topologies.

### 4.3 EVALUATION FROM PRACTICALITY PERSPECTIVES

**Solvability via Offline Evaluation**. Traditional online testing of NFV-RA algorithms, while reflecting real-world scenarios, complicates direct comparisons of algorithmic solvability. The continuous evolution of the PN state, coupled with dynamic VN arrivals and lifetimes, makes it difficult to isolate performance. Specifically, it is challenging to determine if solution failures arise from an algorithm's inherent limitations or from transient unsolvable network states, thereby hindering fair comparisons on solvability. To address this, Virne provides a controlled offline evaluation environment. This features a series of static instances, each comprising a PN and a VN, evaluating algorithms' abilities to find high-quality, feasible solutions under reproducible conditions. Through such evaluation, we assess the fundamental solv-

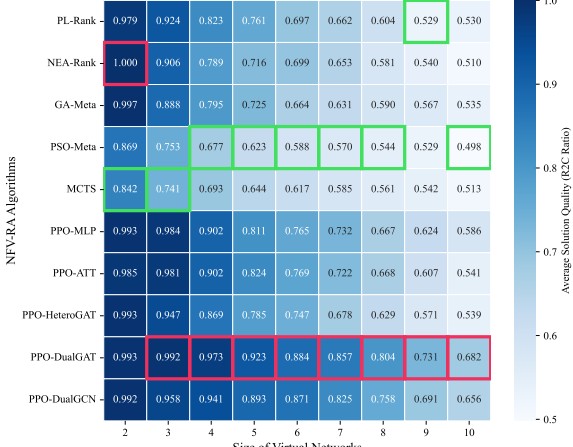

Figure 5: Results on the solvability study. For each size of VN, we highlight the worst-performing solver in green and the best-performing solver in red.

ability of each method, not only across an entire dataset but also for distinct VN scales. This evaluation protocol offers a granular view of algorithmic solvability as VN complexity grows.

Concretely, we evaluate the solution quality, i.e., R2C, achieved by different algorithms for VNs of varying sizes. We conduct experiments on the representative PN, WX100. The results are shown in Figure 5, which presents average solution quality for VN sizes ranging from 2 to 10. There is a generally decreasing trend as the size of the VN increases due to increased complexity. We observe that ❼ *RL methods with advanced graph representations exhibit superior solvability across most VN sizes.* For instance, PPO-DualGAT consistently achieves the highest R2C ratios, particularly for larger VNs. Conversely, simpler RL policies like PPO-MLP perform well on small VNs but degrade noticeably on larger ones due to limited representation ability. This highlights that ❽ *more powerful foundation policies excel in the vast search space of large VNs, dynamically prioritizing the most critical nodes and links for embedding.* Additionally, it is obvious that ❾ *traditional heuristics and meta-heuristics show mixed results and generally underperform compared to the top RL methods, especially for larger VNs.* Particularly, NEA-Rank initially achieves excellent performance for 2-node VNs, but as the VN size increases, it declines and then falls behind the top RL methods.

**Generalization on Network Conditions**. Network conditions are inherently complex and subject to continuous changes, such as fluctuations in request frequencies and varying resource demands. As such, evaluating the generalization of these trained NFV-RA policies is critical to ensure they can adapt effectively to different, evolving network environments. To address this, we conduct the experiments in various network conditions to study the generalization to conditions of pretrained models, including *(a) Evaluation on Varying Traffic Rates* and *(b) Evaluation on Fluctuating Demand Distribution.* See Appendix D.2 for detailed experimental setup and result analysis.

**Scalability on Network Sizes**. Network systems are growing in size as the physical infrastructure extends and service requests increase. Thus, assessing the scalability of NFV-RA policies is essential to ensure they can maintain their effectiveness as the network size and complexity increase. To assess this, Virne supports scalability evaluations through two primary perspectives: *(a) Performance on Large-scaled Networks* and *(b) analysis of Solving Time Scale* as problem size grows. Detailed experimental setups and results for these evaluations are placed in Appendix D.3.

### 4.4 VALIDATION ON EMERGING NETWORKS

NFV is widely applied across diverse modern networks, where specific scenarios often present unique characteristics. To further validate the adaptability of NFV-RA algorithms in emerging networks, we conduct evaluations in two key environments: *(a) heterogeneous resourcing networks*, which involve varied resource types, and *(b) latency-aware edge networks*, where delay constraints should be satisfied. See Appendix D.4 for experimental analysis.

### 4.5 DISCUSSION ON FUTURE DIRECTION

While RL-based methods for NFV-RA show promise, there are still practical challenges, as observed through our empirical observation. To advance deep RL for NFV-RA, we identify several future directions that also pose significant challenges to existing ML methods. They are included but not limited to (a) developing sophisticated representation learning for cross-graph statuses and attributed constraints, (b) achieving generalizable policies adaptable to varying network scales and dynamic conditions, (c) creating robust learning frameworks to enable learning in conflicting operational constraints, and (d) engineering highly scalable algorithms for extremely large-scale physical infrastructures. We also provide experimental analysis of emerging solutions along these directions, such as safe RL and meta RL-based methods. See Appendix E and Figure 13 for details.

### 5 CONCLUSION

In this paper, we introduce **Virne**, a comprehensive and unified benchmarking framework specifically designed for evaluating deep RL-based algorithms for NFV-RA problem. Virne supports diverse network scenarios, including cloud, edge, and 5G environments, facilitated by customizable simulations. In addition, we provide a modular and extensible pipeline that integrates over 30 diverse algorithms, particularly deep RL-based methods, enabling extensible implementation. Beyond traditional effectiveness metrics, Virne offers practical evaluation perspectives such as solvability, generalization, and scalability. Furthermore, we conduct extensive experiments to evaluate the deep RL-based NFV-RA method from comprehensive perspectives. We provide crucial insights on the impact of implementation techniques and algorithm performance trade-offs. Our extensive empirical analysis also reveals valuable findings on potential challenges of RL-based NFV-RA methods, guiding future research on the interaction of data-driven networking optimization and applied machine learning.

## ETHICS STATEMENT

This work introduces Virne, a benchmarking framework designed to accelerate scientific research in ML for network optimization. This study does not involve human subjects or sensitive data. All datasets are either synthetically generated or based on publicly available network topologies from SNDlib. As such, we identify no direct negative societal impacts from this research.

## REPRODUCIBILITY STATEMENT

To ensure reproducibility, we have made our benchmark and all necessary resources publicly available. The complete source code for the Virne framework, including all algorithm implementations and simulation environments, is provided in the anonymous codebase link. We provide a detailed description of the experimental setup in Section 4.1 and Appendix D. These details consist of implementation details, network topologies, and data generation. Overall, these resources ensure the full reproducibility of our work and empower the community to build upon our findings.

## ACKNOWLEDGMENT

This work was supported in part by the National Key R&D Program of China (Grant No.2023YFF0725001), in part by the National Natural Science Foundation of China (Grant No.92370204), in part by the Guangdong Basic and Applied Basic Research Foundation (Grant No.2023B1515120057), in part by the Key-Area Special Project of Guangdong Provincial Ordinary Universities (2024ZDZX1007).

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

APPENDIX

## A NFV-RA PROBLEM DEFINITIONS

### A.1 BASIC FORMULATION OF NFV-RA

Here, we provide the mathematical formulation of the basic cost minimization model, including constraints and the objective.

### A.1.1 CONSTRAINT CONDITIONS

During the embedding process of a VN $\mathcal{G}_v$ onto the PN $\mathcal{G}_p$, we need to decide two types of boolean variables: (1) $x_i^m = 1$ if virtual node $n_v^m$ is placed in physical node $n_p^i$, and 0 otherwise; (2) $y_{i,j}^{m,w} = 1$ if virtual link $l_{m,w}^v = (n_v^m, n_v^w)$ traverses physical link $l_{i,j}^p = (n_p^i, n_p^j)$, and 0 otherwise. Here, $m$ and $w$ are identifiers for virtual nodes, while $i$ and $j$ are identifiers for physical nodes. A VN request is successfully embedded if a feasible mapping solution is found, satisfying the following constraints:

$$\sum_{n_p^i \in N_p} x_i^m = 1, \forall n_v^m \in \mathcal{N}_v, \tag{2}$$

$$\sum_{n_v^m \in \mathcal{N}_v} x_i^m \leq 1, \forall n_p^i \in \mathcal{N}_p, \tag{3}$$

$$x_i^m C(n_v^m) \leq C(n_p^i), \forall n_v^m \in \mathcal{N}_v, n_p^i \in \mathcal{N}_p, \tag{4}$$

$$\sum_{n_p^i \in \Omega(n_p^k)} y_{k,j}^{m,w} - \sum_{n_p^j \in \Omega(n_p^k)} y_{i,k}^{m,w} = x_k^m - x_k^w, \forall l_{m,w}^v \in \mathcal{L}_v, n_v^k \in \mathcal{N}_p, \tag{5}$$

$$y_{i,j}^{m,w} + y_{j,w}^{m,w} \leq 1, \forall l_{m,w}^v \in \mathcal{L}_v, l_{i,j}^p \in \mathcal{L}_p, \tag{6}$$

$$\sum_{l_{m,w}^v \in \mathcal{L}_v} (y_{i,j}^{m,w} + y_{j,i}^{m,w}) B(l_{m,w}^v) \leq B((l_{i,j}^p)), \forall (l_{i,j}^p) \in \mathcal{L}_p. \tag{7}$$

Here, $\Omega(n_p^k)$ denotes the neighbors of $n_p^k$. Constraint (2) ensures that every virtual node is mapped to one and only one physical node. Conversely, constraint (3) limits each physical node to hosting at most one virtual node, thus enforcing a unique mapping relationship. This one-to-one mapping constraint enforces intra-request node disjointness, a standard requirement in virtual network embedding to ensure survivability and prevent single points of failure (Fischer et al., 2013). Note that this does not preclude inter-request multi-tenancy; virtual nodes from different VNs can share the same physical node, provided resources allow. Constraint (4) verifies that virtual nodes are allocated to physical nodes with adequate resources. Following the principle of flow conservation, constraint (5) guarantees that each virtual link $(n_v^m, n_v^w)$ is routed along a physical path from $n_p^i$ (the physical node where $n_v^m$ is placed) to $n_p^j$ (the physical node where $n_v^w$ is placed). Constraint (6) eliminates the possibility of routing loops, thereby ensuring that virtual links are routed acyclically. Lastly, constraint (7) ensures that the bandwidth usage on each physical link remains within its available capacity. Overall, constraints (2,3,4) enforce the one-to-one placement and computing resource availability required in the node mapping process. And constraints (5,6,7) ensure the path connectivity and bandwidth resource availability required in the link mapping process.

### A.1.2 OPTIMIZATION OBJECTIVE

Revenue-to-Consumption (R2C) ratio is a widely used metric to measure the quality of solution $S = f_{\mathcal{G}}(I)$. The objective function is maximize the resource utilization as follows:

$$\max \text{R2C}(S) = \varkappa \cdot (\text{REV}(S) / \text{COST}(S)), \tag{8}$$

where $\varkappa$ is a binary variable indicating the solution's feasibility; $\varkappa = 1$ for a feasible solution and $\varkappa = 0$ otherwise. When the solution is feasible, $\text{REV}(S)$ represents the revenue from the VN, calculated as $\sum_{n_v \in \mathcal{N}_v} C(n_v) + \sum_{l_v \in \mathcal{L}_v} B(l_v)$. If $\varkappa = 1$, $\text{COST}(S)$ denotes the resource consumption of PN, calculated as $\sum_{n_v \in \mathcal{N}_v} C(n_v) + \sum_{l_v \in \mathcal{L}_v} (|f_{\mathcal{L}}(l_v)| \times B(l_v))$. Here, $|f_{\mathcal{L}}(l_v)|$ quantifies the length of the physical path $\rho_p$ routing the virtual link $l_v$.

## A.2 Extensions in Emerging Networks

The application of NFV is pivotal in emerging network scenarios, which require additional considerations of their unique challenges. Particularly, we discuss several extended definitions of NFV-RA in popular scenarios to better align with their practical requirements.

**Resource Heterogeneity**    In modern data center networks, computing resources are often heterogeneous, meaning physical nodes may have varying capacities in terms of CPU, GPU, memory, etc. To account for this, we extend the basic model by incorporating heterogeneous resources, where both virtual and physical nodes are associated with a set of computing resources $\mathcal{C}$. Thus, a physical node $n_p$ for placing virtual node $n_v$ must have enough resources across all types, i.e., $C(n_v) \leq C(n_p), \forall n_v \in \mathcal{N}_v, C \in \mathcal{C}, n_p = f_{\mathcal{N}}(n_v)$.

**Latency Requirement**    In time-sensitive networks (e.g., edge computing and 5G), satisfying latency requirements is crucial. We consider the latency requirement of virtual link $l_v$ as $D(l_v)$ and the incurred latency of physical link $p_v$ as $D(l_p)$. The latency of physical path $\rho_p$ that routes $l_v$ should not exceed such specified threshold, i.e., $D(\rho_p) \leq D(l_v)$, where $D(\rho_p) = \sum_{l_p \in \rho_p} D(l_p)$.

**Energy Efficiency**    Energy consumption is a significant concern in green data centers due to high sustainability and economic efficiency. Energy-efficient NFV-RA also considers the energy consumption minimization of physical infrastructure. We denote the energy consumed by the physical node $n_p$ as $E(n_p)$, associated with its status (idle or active) and workload. The objective function is to optimize both resource utilization and energy consumption, formulated as: $\max -w_a \cdot \sum_{n_p \in \mathcal{N}_p} E(n_p) + w_b \cdot \text{R2C}(S)$, where $w_a$ and $w_b$ are weights of different objectives.

Note that these definitions extend the basic concepts of modeling, constraints, and objectives. Variations of NFV-RA for other scenarios can be easily derived using the approaches discussed above.

## B  Related Work

As network scenarios become increasingly diverse and complex (e.g., cloud, edge, 5G), NFV has emerged as a critical paradigm for flexible service provisioning. Efficient resource management in NFV spans the entire service lifecycle, encompassing tasks such as efficient allocation (Jin et al., 2020), adaptive scaling (Fei et al., 2018), and dynamic migration (Zhang et al., 2021b). Among these, Resource Allocation in NFV (NFV-RA) forms the foundation. It is typically formulated as an embedding or placement problem, with the central challenge of mapping virtual requirements onto physical infrastructure. While many studies investigate NFV-RA in emerging networks while addressing additional factors (e.g., latency, energy, heterogeneity), their formulations often follow a similar methodological backbone. This shared structure enables a unified perspective on how NFV-RA techniques have evolved over time. In this section, we focus on the methodological aspects and review the development of key approaches in NFV-RA. Then, we describe existing benchmarks to emphasize the gap between Virne. Finally, we compare NFV-RA with other COPs, highlighting the need for specific designs.

### B.1  Traditional NFV-RA Algorithms

Earlier studies for the NFV-RA problem employed *exact solvers*, such as integer linear programming (ILP)(Shahriar et al., 2018) and mixed integer programming (MIP)(Chowdhury et al., 2009), to find optimal solutions. However, their high computational complexity makes them impractical for real-world scenarios, especially in larger networks where dynamic service requests require rapid solutions. To address these limitations, *heuristic-based methods* emerged as alternatives, offering suboptimal yet computationally efficient solutions. Among these, node-ranking strategies stand out as a prominent approach (Gong et al., 2014; Zhang et al., 2018; Fan et al., 2021; 2023). These strategies construct solutions by prioritizing the virtual and physical nodes based on specific metrics to guide the node mapping process, followed by link mapping. For instance, Node Resource Management (NRM) (Zhang et al., 2018) ranks nodes by weighting both node and link resources, while the Node Essentiality Assessment (NEA) (Fan et al., 2023) incorporates topology connectivity into decision-making. Although reducing computational overhead, they rely heavily on manual design

Table 4: Summary of existing studies on RL-based NFV-RA algorithms. (a) Existing studies investigate RL-based NFV-RA algorithms in various network scenarios, such as cloud computing and latency-aware edge computing. (b) These studies employ different RL methods, such as DQN and PPO, to optimize customized neural policies, including CNN- or GNN-based architectures. (c) They also vary in implementation techniques, including whether they incorporate intermediate rewards in the reward function, use action masking mechanisms to prevent selecting nodes with insufficient resources, or leverage topological features as augmented inputs. (d) Additionally, we summarize the benchmarks used in these studies, noting whether they rely on existing benchmarks or use custom ones that are not publicly available.

| | Network Scenario | | Core RL Framework | | Implementation Techniques | | | Code Base |
|---|---|---|---|---|---|---|---|---|
| | Network System | Additional Considerations | Training Method | Neural Policy | Intermediate Rewards | Action Masking | Topological Features | Used Benchmark |
| (He et al., 2023) | Edge | / | A2C | LSTM, Attention | ✓ | ✗ | ✗ | Not available |
| (Zhang et al., 2024a) | Edge | Energy, Latency | PG | CNN | ✗ | ✓ | ✓ | Not available |
| (Tan et al., 2024) | 5G Slicing | Security | DQN | Heterogeneous GCN | ✓ | ✗ | ✗ | Not available |
| (Liu et al., 2020) | 5G Slicing | Latency | PPO | MLP | ✗ | ✓ | ✗ | Not available |
| (Irawan et al., 2023) | 5G Slicing | Latency | PPO | GCN | ✗ | ✓ | ✗ | **Virne** |
| (Tian et al., 2024) | Internet of Things | Latency | PPO | GNN | ✓ | ✓ | ✓ | Not available |
| (Fu et al., 2020) | Internet of Things | Latency | DQN | CNN | ✓ | ✗ | ✗ | Not available |
| (Guo et al., 2022) | Internet of Things | - | A3C | MLP | ✗ | ✗ | ✗ | Not available |
| (Maity et al., 2024) | Satellite | Latency | DQN | MLP | ✗ | ✗ | ✗ | Not available |
| (Zhang et al., 2023) | Data Plane | Latency | A3C | GCN | ✓ | ✓ | ✓ | Not available |
| (Zhang et al., 2024b) | Space-Air-Ground | Latency | PG | CNN | ✗ | ✗ | ✗ | Not available |
| (Haeri & Trajković, 2017) | Cloud | / | MCTS | / | ✗ | ✗ | ✗ | VNE-Sim |
| (Dolati et al., 2019) | Cloud | / | PG | CNN | ✓ | ✗ | ✗ | Not available |
| (Xiao et al., 2019) | Cloud | Latency | PG | MLP | ✓ | ✗ | ✗ | Self-implemented |
| (Solozabal et al., 2020) | Cloud | Energy | PG | LSTM; Attention | ✓ | ✗ | ✗ | Self-implemented |
| (Yao et al., 2020) | Cloud | / | PG | RNN | ✗ | ✗ | ✗ | Not available |
| (Yan et al., 2020) | Cloud | / | A3C | GCN | ✓ | ✓ | ✓ | VNE-Sim |
| (Huang et al., 2021) | Cloud | / | DQN | MLP | ✓ | ✗ | ✗ | VNE-Sim |
| (Wang et al., 2021) | Cloud | / | A3C | GCN, GRU | ✗ | ✓ | ✓ | **Virne** |
| (Zhang et al., 2021a) | Cloud | Security | PG | CNN | ✓ | ✗ | ✓ | Not available |
| (Ma et al., 2023) | Cloud | / | PG | GCN | ✗ | ✗ | ✗ | Not available |
| (Zhang et al., 2022) | Cloud | / | A3C | GCN, GRU | ✓ | ✓ | ✗ | VNE-Sim |
| (Geng et al., 2023) | Cloud | / | PPO | GNN | ✓ | ✓ | ✓ | **Virne** |
| (Xiao, 2023) | Cloud | / | PG | MLP | ✓ | ✓ | ✓ | Not available |
| (Wang et al., 2023) | Cloud | / | PPO | Edge-aware GAT | ✓ | ✓ | ✓ | **Virne** |
| (Sahraoui et al., 2024) | Cloud | Energy | A2C | Attention | ✓ | ✓ | ✗ | **Virne** |
| (Wang et al., 2024c) | Cloud | Heterogeneity | PPO | Cross-GCN | ✓ | ✓ | ✓ | **Virne** |
| (Wang et al., 2025b) | Cloud, Edge | Latency | PPO | Heterogeneous GAT | ✗ | ✓ | ✓ | **Virne** |

with domain-specific knowledge and lack adaptation to diverse scenarios, which can lead to inferior results in complex NFV-RA requirements. Furthermore, *meta-heuristics* have been adopted to solve NFV-RA by leveraging the evolutionary process (Dehury & Sahoo, 2019; Leivadeas et al., 2013; Fajjari et al., 2011). These algorithms, such as Genetic Algorithms (GA) (Zhang et al., 2019; Dehury & Sahoo, 2019; Wang et al., 2025a) and Particle Swarm Optimization (PSO) (Su et al., 2014; Jiang & Zhang, 2021), explore the solution space by iteratively evolving a population of candidate solutions. However, they are computationally expensive and hyperparameter-sensitive, limiting their practicality.

## B.2 RL-BASED NFV-RA ALGORITHMS

In recent years, machine learning (ML)-based methods have gained prominence in solving NFV-RA problems due to their superior performance and adaptability to dynamic network conditions (Wu et al., 2024a). Several works (Blenk et al., 2016; Thakkar et al., 2020) have explored predictive models trained on high-quality datasets to forecast the future service requests. However, acquiring labeled, high-quality datasets for large-scale and unseen network scenarios remains impractical, limiting their applicability. More dominantly, RL has demonstrated its promise for NFV-RA tasks (Haeri & Trajković, 2017; Xiao et al., 2019; Yao et al., 2020; Wang et al., 2021; Yan et al., 2020; Zhang et al., 2022; Dolati et al., 2019; Wang et al., 2023; Geng et al., 2023; Wang et al., 2024c; Gao et al., 2025), mainly due to its label-free nature and adaptation to handle dynamics of network. RL-based methods model NFV-RA as a Markov Decision Process (MDP), allowing an agent to learn optimal policies through iterative interactions with the environment. In general, we unified existing RL-based methods for NFV-RA under a framework with three core components: MDP modeling, policy architecture, and training methods. These methods conceptualize the node mapping allocation process as a sequential decision-making task, where physical nodes are incrementally selected to host VNFs. To execute these decisions, various policy network architectures are designed to represent the network state and generate corresponding actions. Then, they leverage a specific RL method to optimize the policy network with collected experience during interactions. For example, (Xiao et al., 2019) utilized a multilayer perceptron (MLP) and trained it using the policy gradient (PG) algorithm, while the work (Yan et al., 2020) combined MLPs with graph convolutional networks (GCNs) Wang et al. (2024b); Deng et al. (2024) and employed the asynchronous advantage actor-critic (A3C) algorithm. While RL-based methods offer notable advantages in terms of performance and adaptability, there is a lack of in-depth analysis of the impact of specific model design choices and consistent performance comparison. Moreover, systematically identifying the remaining significant challenges in RL-based NFV-RA is critical to open opportunities for future exploration and innovation.

## B.3 EXISTING NFV-RA BENCHMARKS

For the NFV-RA problem, several benchmarks have been developed for simulation and evaluation. As summarized in Table 1, the most notable existing benchmarks include: VNE-Sim[1] (Haeri & Trajković, 2016), ALEVIN [2] (Beck et al., 2014), ALib[3] (Beck et al., 2014), SFCSim[4] (Xu et al., 2022), and Iflye[5] (Tomaszek, 2021). For instance, VNESim(Haeri & Trajković, 2016) supports three heuristic algorithms and focuses exclusively on cloud-based simulations, evaluating solutions based on their effectiveness. Similarly, SFCSim (Xu et al., 2022), incorporates three additional heuristic methods, continuing the trend of focusing on cloud-based scenarios However, most of these benchmarks are limited to cloud-based simulations and lack the flexibility to accommodate more modern network scenarios, such as edge computing or 5G environments. Moreover, these benchmarks generally implement only a few exact and heuristic algorithms and focus mainly on effectiveness evaluation. These limitations highlight the need for more comprehensive benchmarks that can assess algorithms across a broader range of network environments and evaluation perspectives.

---

[1]https://tehreemf.wixsite.com/vne-sim
[2]https://sourceforge.net/p/alevin/wiki/home/
[3]https://github.com/vnep-approx/alib
[4]https://github.com/SFCSim/SFCSim
[5]https://github.com/Echtzeitsysteme/iflye

### B.4    RL FOR COMBINATORIAL OPTIMIZATION

RL has gained significant attention for solving Combinatorial Optimization Problems (COPs), where the goal is to learn high-quality solutions (Bengio et al., 2021). These problems span various domains, including the Traveling Salesman Problem (TSP) (Kool et al., 2018), Vehicle Routing Problem (VRP) (Zhou et al., 2023), and bin packing (Zhao et al., 2021). RL-based approaches to solving these problems generally fall into two categories: construction methods that build a solution incrementally from scratch and improvement methods that start with an initial solution and iteratively refine it. Compared with these classic COPs, the NFV-RA problem presents unique challenges, highlighting the need for specialization of RL approach design. First, NFV-RA requires real-time decision-making to meet strict network service requirements. NFV-RA algorithms must execute decisions almost immediately, making construction methods more suitable to incrementally build solutions within time constraints. Second, NFV-RA operates in highly dynamic environments where both service demands and network resources fluctuate in real time. This variability requires RL methods that can adapt to changes in network topologies, resource availability, and demand patterns. Lastly, the state space of NFV-RA is highly complex, involving intricate interactions such as cross-graph mapping and bandwidth-constrained routing. These factors require RL methods to navigate a multidimensional decision space, accounting for diverse resource types and connectivity requirements.

## C    BENCHMARK DETAILS

### C.1    SIMULATION CONFIGURATION

Virne offers a highly customizable simulation framework designed to accommodate the diverse and complex nature of modern network environments. As illustrated in Figure 6, users can define the network scenarios and conditions within the simulator through configuration files. These configurations cover two key components: PN settings and VN requests, providing users the flexibility to simulate a variety of network scenarios.

**Network Topologies**    Virne provides a range of customizable network topologies to simulate different network conditions. Users can select between: *(a) synthetic topologies* that are generated using algorithms such as Waxman and FatTree, which provide structured and predictable network designs for controlled testing; and *(b) realistic topologies* that are sourced from SDNLib and the Topology Zoo, offering more realistic and complex network structures that mimic real-world environments. Users can specify both the PN and VN topologies through their configuration files, allowing the reflection of various network scales and topological characteristics.

**Resource Availability**    Virne supports the customization of resource availability across multiple levels of the network, including nodes, links, and graph. This flexibility enables simulations that represent different resource conditions, such as: *(a) resource types*: Users can define various types of resources, including computing resources (e.g., CPU, GPU, memory) and network resources (e.g., bandwidth). This allows for the modeling of diverse resource requirements in both PN and VN settings; *(b) availability distribution*: The resources can be distributed across the network using different statistical models, such as uniform or exponential distributions, which reflect real-world variations in resource supply and demand. This level of customization provides users with the ability to model networks with differing levels of resource availability.

**Service Requirements**    Virne allows users to define specific service requirements, which add complexity and realism to the simulations. These requirements can be tailored to address a wide range of network use cases, reflecting varying demands and constraints. As illustrated in Figure 6, we introduce three key types of service awareness that can be customized:

- *Heterogeneous resources*. To configure this, users can add two new types of node resources (e.g., CPU, GPU) into the *node_attrs_setting* section of the PN and VN configuration files.
- *Latency constraints*. By introducing the latency into the *link_attrs_setting* section of the PN and VN configuration files, users can specify maximum latency thresholds for virtual links and the physical paths used for routing.

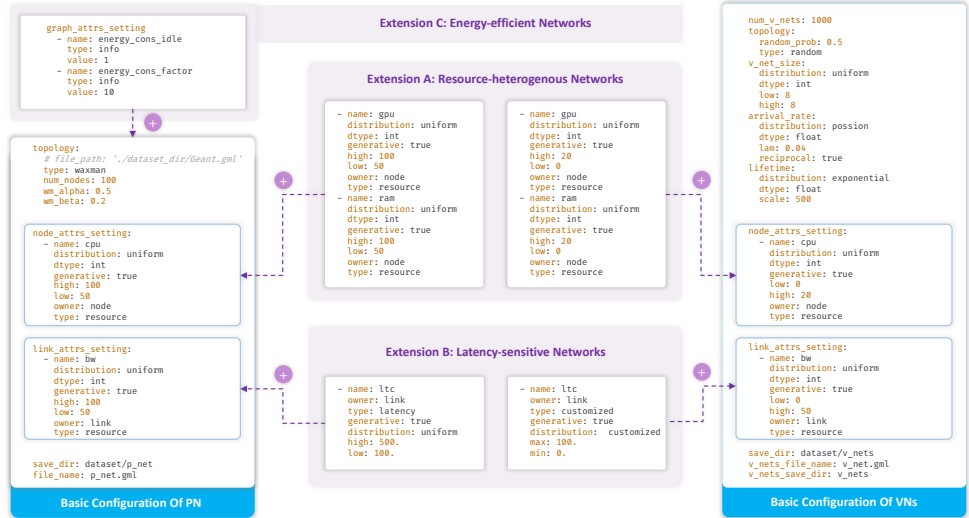

Figure 6: An example of basic configurations on both PN and VNs, along with their extensions. By adding specific settings on the levels of node, link, or graph, Virne can be easily extended to support emerging networks with additional awareness.

- *Energy efficiency*. By including energy consumption parameters into the *graph_attrs_setting* section of the PN configuration files, users can simulate green data centers and optimize resource allocation with sustainability in mind.

Through these service requirements, Virne enables the simulation of networks with diverse operational demands. Moreover, Virne supports adaptable constraint modeling to accommodate specific scenarios. For instance, users can enable VNF aggregation strategies (e.g., for SFC) by relaxing node disjointness constraints via the `reusable` parameter. This provides users with the flexibility to model complex and dynamic network systems beyond standard definitions.

## C.2 MDP VARIATION FOR NFV-RA

Apart from the MDP formulations introduced in Section 3.3.1, several extended MDPs for NFV-RA are also proposed to handle the specific aspects of NFV-RA problems.

### C.2.1 MULTI-TASK MDP FOR META RL METHODS

In practical network systems, VNRs often exhibit significant diversity in their characteristics, such as varying sizes, topologies, and resource demands. A standard single-policy approach struggles to generalize across this wide spectrum. To address this, a multi-task MDP formulation can be employed, framing the embedding of different categories of VNRs as distinct tasks. Specifically, VNRs are grouped into tasks $\mathcal{M}_i$ drawn from a distribution $p(\mathcal{M})$. The objective is to train a **meta-policy** $\pi_\phi$ that captures common, transferable strategic knowledge across all tasks. This meta-policy can then be rapidly adapted with minimal data to derive specialized sub-policies $\pi_{\theta_i}$ for each task. The meta-optimization objective is to find meta-parameters $\phi$ that maximize the expected performance over the task distribution after adaptation:

$$\max_\phi \mathbb{E}_{\mathcal{M}_i \sim p(\mathcal{M})} \left[ J(\theta_i) \right] = \mathbb{E}_{\mathcal{M}_i \sim p(\mathcal{M})} \left[ J(f_\phi(\mathcal{D}_i)) \right] \tag{9}$$

where $J(\theta_i)$ is the expected return for the policy $\pi_{\theta_i}$ on task $\mathcal{M}_i$, and $\theta_i = f_\phi(\mathcal{D}_i)$ represents the parameters of the sub-policy adapted from the meta-policy $\phi$ using a small amount of task-specific experience $\mathcal{D}_i$. The meta-policy is then updated by aggregating gradients from the adapted policies, for instance, using a second-order meta-gradient over the task-specific losses.

### C.2.2 CONSTRAINED MDP FOR SAFE RL METHODS

The NFV-RA problem is fundamentally defined by its hard resource constraints, where any violation renders a solution infeasible. Standard MDPs, which typically integrate constraint violations into the

reward signal via simple penalties or doing nothing, often fail to guarantee zero-violation solutions. A more rigorous approach is to formulate the problem as a Constrained CMDP. This method explicitly separates the primary reward from costs associated with constraint violations and is formally expressed as a tuple $\langle S, A, P, R, H, C, \gamma \rangle$. Here, in addition to standard MDP components, $H$ is a violation function that measures the degree of constraint violation at each state, and $C$ is a cost function that maps violations to a non-negative cost, i.e., $C(s) = \max(H(s), 0)$. The objective is to learn a policy $\pi$ that maximizes the expected cumulative reward $J_r(\pi)$ while ensuring the expected cumulative cost $J_c(\pi)$ remains below a predefined threshold $d$ (e.g., $d = 0$):

$$
\begin{aligned}
\max_{\pi} \quad & J_r(\pi) = \mathbb{E}_{\tau \sim \pi} \left[ \sum_{t=0}^{T} \gamma^t R(s_t, a_t) \right] \\
\text{s.t.} \quad & J_c(\pi) = \mathbb{E}_{\tau \sim \pi} \left[ \sum_{t=0}^{T} \gamma^t C(s_t) \right] \leq d
\end{aligned}
\tag{10}
$$

To enforce stricter state-wise safety, this model can be enhanced with reachability analysis. This extension aims to ensure the policy remains within a feasible region where future constraint violations are avoidable, often by optimizing an objective that guarantees the worst-case long-term violation, or feasible value function $V_h^\pi(s)$, remains non-positive.

### C.3 IMPLEMENTED NFV-RA ALGORITHMS

#### C.3.1 RL-BASED ALGORITHM IMPLEMENTATIONS

Our RL-based implementations are structured around core components: RL training algorithms that define the learning paradigm, neural policy architectures that parameterize the agent's decision-making function, and additional techniques that can enhance learning or address specific challenges in NFV-RA. This modularity allows researchers to easily experiment with different combinations and contribute new components.

**RL training Methods**  Virne supports several foundational and advanced RL training algorithms, which are used to guide the learning process of the neural policies.

- *Monte Carlo Tree Search (MCTS) (Haeri & Trajković, 2017)* is a planning algorithm that explores the decision space by building a search tree, balancing exploration and exploitation to find action sequences.

- *Policy Gradient (PG or REINFORCE) (Sutton et al., 1999)* is a policy-based RL method that learns a parameterized policy to maximize expected returns, i.e., solution quality.

- *Asynchronous Advantage Actor-Critic (A3C) (Mnih et al., 2016)* is an actor-critic-based RL algorithm that uses multiple parallel actors, each interacting with its own copy of the environment. It learns both a policy (actor) and a value function (critic) to estimate the advantage of taking certain actions, leading to more efficient learning than pure PG methods.

- *Proximal Policy Optimization (PPO) (Schulman et al., 2017)* is a popular RL method known for its stability and strong empirical performance across a wide range of tasks. It achieves policy optimization stability by using a clipped surrogate objective.

- *Deep Q-Network (DQN) and variants (Mnih et al., 2013)* are value-based RL methods that aim to learn the state-action value space. However, NFV-RA often involves large, structured action spaces not directly amenable to these methods.

**Neural Policy Architectures**  The neural policy architecture defines how the RL agent perceives the environment (state representation) and decides on actions. Virne implements a range of architectures to capture attribute and structural information in both the PN and VN.

- *Multi-Layer Perceptron (MLP)-based Policy (Liu et al., 2020; Maity et al., 2024; Xiao, 2023; Xiao et al., 2019).* It concatenates the features of the current to-be-placed virtual node with every physical node's features. Then, it fed them into multiple fully connected layers and outputs probabilities for selecting each physical node for placement.

- *Convolutional Neural Network (CNN)-based Policy* (Zhang et al., 2024a; Fu et al., 2020; Zhang et al., 2024b; 2021a; Dolati et al., 2019). They use CNN to process graph-structured data by treating node features and adjacency information as grid-like inputs. Similar to the MLP, virtual link demands are integrated as node features.

- *Attention-based Policy*(He et al., 2023; Solozabal et al., 2020). It first embeds the features of the current to-be-placed virtual node to form a query. Then, the features of each physical node are similarly embedded to form keys and values. The attention mechanism then computes a weighted sum of physical node values based on their compatibility with the virtual node query.

- *Graph Convolutional Network (GCN)-based policy (Zhang et al., 2023; Ma et al., 2023; Yan et al., 2020)* . It uses GCN to learn node representations by aggregating information from their local neighborhoods. In Virne, GCNs are used to process the PN topology, often with virtual node demands incorporated as additional node features on the PN nodes.

- *Graph Convolutional Network (GCN)-based policy*. It uses GCN to learn node representations by aggregating information from their local neighborhoods. GCNs is mainly used to process the PN topology, often with virtual node demands incorporated as additional node features on the PN nodes.

- *GCN & Sequence-to-Sequence (GCN&S2S) Policy (Wang et al., 2021).* This hybrid architecture combines GCNs for graph representation learning with a Sequence-to-Sequence (S2S) model (Sutskever et al., 2014), an RNN-based encoder-decoder with attention, to generate an ordered sequence of actions.

- *GAT&S2S Policy (Wang et al., 2023).* Similar to GCN&S2S, but replaces the GCN encoder(s) with GAT(s).

- *Graph Attention Network (GAT)-based policy* (Wang et al., 2023). GATs extend GCNs by incorporating attention mechanisms into the neighborhood aggregation process. Particularly, it also incorporates edge features (e.g., link bandwidth, latency) into the attention calculation.

- *Dual GCN (DualGCN)-based policy (Wang et al., 2024c).* This architecture uses two separate GCNs, i.e., one to embed the topology and features of the VN, and another for the PN. The embeddings from both graphs are then combined into an MLP to make placement.

- *Dual GAT (DualGAT)-based Policy (Wang et al., 2024c)*: Similar to DualGCN, but it utilizes GATs for both VN and PN embedding.

- *Heterogeneous Graph Attention Network (HeteroGAT)-based Policy (Tan et al., 2024; Wang et al., 2025b).* Recent studies model NFV-RA instances as a heterogeneous graph, where virtual nodes, physical nodes, virtual links, and physical links are different types of nodes/edges with distinct features. Additionally, HeteroGAT introduces the cross-graph connection links to represent the historical mapping status, i.e, if a virtual node is placed onto a physical node, a heterogeneous link will be created.

**Additional Implementation Techniques**

- *Reward Function Design*. NFV-RA is characterized by its combinatorial space and complex constraints for NFV-RA, which may present a challenging sparse reward problem. Virne allows exploration of different reward structures.
    - *No Intermediate Reward (NoIR)*. The agent only receives a terminal reward based on the final $R2C(S)$ if the entire VN is successfully embedded, and 0 otherwise.
    - *Fixed Intermediate Reward (FIR)*. The agent receives a small positive fixed reward (ImR_value) for each successful virtual node placement and link routing step. A negative reward is given for failed steps. The final $R2C(S)$ is added to the sum of intermediate rewards if the overall embedding is successful.
    - *Adaptive Intermediate Reward (AIR)*. A specialized version of FIR considers $\frac{1}{|\mathcal{N}_v|}$ as an intermediate reward value, where $|\mathcal{N}_v|$ is the number of virtual nodes in the VN. This normalizes the scale of intermediate rewards based on the complexity of the VN.

- *Feature Engineering Combinations*. Raw node/link attributes and basic topology might not be sufficient for optimal RL performance. Some studies use engineered features to augment the input to neural policies. These include:

Table 5: Implemented Traditional NFV-RA Algorithms

| | Core Strategy |
|---|---|
| MIP (Chowdhury et al., 2009) | Mixed-Integer Programming (MIP) |
| R-Rounding (Chowdhury et al., 2012) | Random Rounding |
| D-Rounding (Chowdhury et al., 2012) | Deterministic Rounding |
| RW-Rank (Cheng et al., 2011) | Random Walk (RW) |
| GRC-Rank (Gong et al., 2014) | Global Resource Control (GRC) |
| NRM-Rank (Zhang et al., 2018) | Node Resource Management (NRM) |
| NEA-Rank (Fan et al., 2023) | Node Essentiality Assessment (NEA) |
| PL-Rank (Fan et al., 2021) | Priority of Location (RL) |
| GA-Meta (Zhang et al., 2019) | Genetic Algorithm (GA) |
| PSO-Meta (Jiang & Zhang, 2021) | Particle Swarm Optimization (PSO) |
| ACO-Meta (Fajjari et al., 2011) | Ant Colony Optimization (ACO) |
| SA-Meta (Wang et al., 2013) | Simulated Annealing (SA) |
| TS-Meta (Wang et al., 2017) | Tabu Search (TS) |

- – *(a) Node Embedding Status*. Binary flags indicating whether a physical node is currently hosting a virtual node or whether a virtual node has already been placed.
- – *(b) Topological Features*. Standard graph centrality measures computed for both PN nodes and VN nodes, i.e., degree, closeness, betweenness, and eigenvector.

- *Action Masking Mechanism*. During training, not all actions (e.g., placing a virtual node on a physical node) are valid at every step due to resource constraints or additional service requirements. Action masking is a technique where invalid actions are explicitly disallowed by placing the selection probability with zero.

### C.3.2 TRADITIONAL ALGORITHM IMPLEMENTATION

Virne implements 10+ traditional algorithms, which can be broadly categorized into three primary types: exact and rounding methods, node ranking-based approaches, and meta-heuristic methods. We summarize these key algorithms and their core strategies in Table 5. Next, we elaborate on each category and introduce implemented algorithms as follows.

**Exact and rounding methods for NFV-RA** either guarantee optimal solutions or provide approximate solutions through rounding techniques, mainly based on mathematical solvers.

- *MIP* (Chowdhury et al., 2009): Mixed-Integer Programming (MIP) is an exact method that provides optimal solutions by solving a system of linear equations with integer constraints.
- *R-Rounding* (Chowdhury et al., 2012): A rounding method that applies random rounding to generate approximate solutions for NFV-RA.
- *D-Rounding* (Chowdhury et al., 2012): This method uses deterministic rounding to map continuous variables to discrete values while providing an approximation to the optimal solution.

**Node ranking-based methods for NFV-RA** that first prioritize nodes using different strategies for node mapping, then execute the link mapping stage with the shortest path algorithm.

- *NRM-Rank* (Zhang et al., 2018): A heuristic method based on Node Resource Management (NRM) metric. This approach ranks virtual and physical nodes before mapping with a greedy selection method.
- *GRC-Rank* (Gong et al., 2014): This method uses a Global Resource Control (GRC) strategy based on random walks for node ranking. After ranking, the nodes are mapped accordingly.
- *NEA-Rank* (Fan et al., 2023): It ranks nodes using the Node Essentiality Assessment (NEA).
- *RW-Rank* (Cheng et al., 2011): A node ranking approach based on random walks to estimate node priorities in the embedding process.

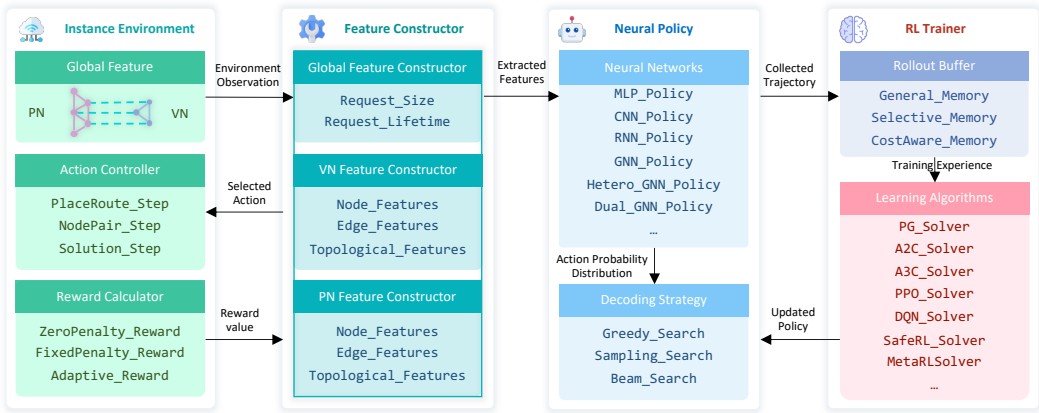

Figure 7: The modular architecture of the RL-based implementation pipeline in Virne. The pipeline is decoupled into three core, interchangeable modules: Instance Environment, RL Agent, and RL Trainer. Each module contains several sub-components (e.g., feature constructor, neural policy, learning algorithm) that can be individually customized and combined.

- *PL-Rank* (Fan et al., 2021): This algorithm considers node proximity and evaluates paths comprehensively, ranking nodes based on their location priorities and the overall efficiency of the physical infrastructure.

**Meta-heuristic methods for NFV-RA** employ nature-inspired processes to iteratively improve solutions. Virne implements the following meta-heuristics for NFV-RA:

- *GA-Meta* (Zhang et al., 2019): A meta-heuristic method based on genetic algorithms. It models each node mapping solution as a chromosome and iteratively explores the solution space by simulating the process of genetic selection and evolution.

- *PSO-Meta* (Su et al., 2014): A meta-heuristic using Particle Swarm Optimization (PSO), where particles explore the NFV-RA solution space by adjusting their positions based on their own and neighbors' experiences.

- *ACO-Meta* (Fajjari et al., 2011): A meta-heuristic based on Ant Colony Optimization (ACO) that simulates ant foraging behavior to explore the NFV-RA solution space and update pheromone trails to guide the search.

- *SA-Meta* (Wang et al., 2013): A meta-heuristic using Simulated Annealing (SA) to approximate the global optimum, exploring the NFV-RA solution space with both upward and downward transitions to escape local minima.

- *TS-Meta* (Wang et al., 2017): A meta-heuristic based on Tabu Search (TS), which uses a memory structure to store visited solutions, promoting diversity and avoiding local optima.

## C.4 MODULAR AND EXTENSIBLE IMPLEMENTATION PIPELINE

To facilitate rapid implementation and community-driven development, Virne is with a highly modular and extensible implementation pipeline for RL-based methods. This design is guided by the principle of separation of concerns, where distinct functional components of the RL workflow can be independently developed, configured, and interchanged. This section outlines the architecture of this pipeline and a customized configuration example, as illustrated in Figures 7 and 8.

### C.4.1 MODULARITY DESCRIPTION OF PIPELINE

As shown in Figure 7, our pipeline consists of the following high-level, interoperable modules.

**Instance Environment** simulates the NFV system and handles the agent's interactions. It is designed with customizable components, including the *Action Controller*, which defines the step-wise logic for

```
# Modular Settings of Instance Environment
action_step_env_cls: BaseEnv = ...              # e.g., JointPRInstanceRLEnv
config.reward_calculator_name: str = ...        # e.g., adaptive

# Modular Settings of Feature Constructor
config.if_use_topological_metrics: bool = ...
config.if_use_node_status_flags: bool = ...

class CustomEnv(action_step_env_cls):
    def __init__(self, p_net, v_net, controller, recorder, counter, logger, config, **kwargs):
        super(CustomEnv, self).__init__(p_net, v_net, controller, recorder, counter, logger,
                            config, **kwargs)

# Modular Settings of Neural Policy
tensor_obseration_func: Callable = ... # e.g., TensorConvertor.obs_as_tensor_for_gnn
policy_builder: Callable = ...         # e.g., PolicyBuilder.build_gcn_mlp_policy

# Modular Setting of RL Trainer
rl_solver_cls: BaseRLSolver = ...      # e.g., PGGcnMlpSolver

class CustomSolver(InstanceAgent, rl_solver_cls):
    def __init__(self, controller, recorder, counter, logger, config, **kwargs):
        InstanceAgent.__init__(self, CustomEnv)
        rl_solver_cls.__init__(self, controller, recorder, counter, logger, config,
                            policy_builder, tensor_obseration_func, **kwargs)
```

Figure 8: A code example of implementing a new NFV-RA algorithm via module customization.

state transition in a gym-style environment; the *Reward Calculator*, which allows for different reward schemes to be easily plugged in.

**Feature Constructor** is responsible for processing the raw state information from the environment (PN and VN) and extracting meaningful features. It supports various feature engineering strategies, such as including basic node/edge attributes or augmenting them with advanced topological metrics.

**Neural Policy** represents the core of the agent's learned strategy. Our pipeline supports a wide array of *Neural Networks*, from simple MLPs to more complex structures like CNNs, RNNs, and various GNNs. The user can also customize the *Decoding Strategy* that translates the probabilistic output of the neural policy into a concrete action.

**RL Trainer** manages the learning process, which includes the rollout buffer that stores the interaction experiences (state, action, reward, etc.) used for training; and *Learning Algorithms:* that is used to update the agent's policy. Virne integrates a comprehensive suite of solvers, such as PG, A2C, PPO, and more advanced options like SafeRL and MetaRL solvers.

### C.4.2 EXTENSIBILITY IN PRACTICE

This modular design enables users to develop new algorithms flexibly, as shown in Figure 8.

**Configuration-driven Customization:** Users can easily assemble different pipelines by modifying configuration files. As shown in the top part of Figure 8, parameters like the reward calculator or the types of features to use can be set with simple string or boolean flags. This enables extensive experimentation without writing any new code.

**Class-based Extension:** For more advanced customization of flexible components, as shown in the bottom part of Figure 8, users can create custom environments or solvers (e.g., CustomEnv, CustomSolver) by inheriting from base classes. New components, such as a novel policy architecture or a custom RL solver, can be injected as callable functions or classes during instantiation. This powerful design pattern significantly lowers the barrier for implementing and testing new research ideas within a standardized framework.

### C.5 EVALUATION METRIC DEFINITIONS

We provide the definitions of key evaluation metrics commonly used to evaluate the NFV-RA algorithms as follows.

- **Request Acceptance Rate (RAC)** measures the proportion of VN requests that the system successfully accepts. Formally, it is given by

$$\text{RAC} = \frac{\sum_{t=0}^{\mathcal{T}} |\tilde{\mathcal{I}}(t)|}{\sum_{t=0}^{\mathcal{T}} |\mathcal{I}(t)|} \times 100, \tag{11}$$

where $\mathcal{T}$ denotes the total operational time of the network system. $\mathcal{I}(t)$ is the set of all VN requests arriving at time slot $t$, and $\tilde{\mathcal{I}}(t)$ is the subset of those requests that are accepted. The notation $|\cdot|$ denotes the cardinality of a set.

- **Long-term Revenue-to-cost Ratio (LRC)** evaluates the economic efficiency of the system by comparing revenue to resource consumption. It is formulated as

$$\text{LRC} = \frac{\sum_{t=0}^{\mathcal{T}} \sum_{I \in \tilde{\mathcal{I}}(t)} \text{REV}(S) \times \varpi}{\sum_{t=0}^{\mathcal{T}} \sum_{I \in \tilde{\mathcal{I}}(t)} \text{COST}(S) \times \varpi} \times 100, \tag{12}$$

where $S = f_G(I)$ as before, and $\text{REV}(S)$ and $\text{COST}(S)$ denote the revenue obtained and resources consumed by the embedding, respectively.

- **Long-term Average Revenue (LAR)** quantifies the total revenue generated over a period, reflecting the economic benefits of processing VN requests. It is defined as

$$\text{LAR} = \frac{1}{\mathcal{T}} \sum_{t=0}^{\mathcal{T}} \sum_{I \in \tilde{\mathcal{I}}(t)} \text{REV}(S) \times \varpi, \tag{13}$$

where $E = f_G(I)$ represents the embedded VN request, and $\varpi$ is the lifetime of the corresponding VN.

- **Average Solving Time (AST)** indicates the efficiency of the NFV-RA algorithm by measuring the average consumed time (in seconds) for solving one simulation run.

## C.6 EVALUATION PROTOCOL DEFINITIONS

To provide a rigorous basis for comparing algorithmic performance across diverse NFV scenarios, we formally define the three core evaluation perspectives as follows:

- **Solvability ($\mathcal{S}$):** This metric quantifies an algorithm's ability to satisfy the hard constraints of the NFV problem (e.g., CPU, bandwidth, and latency limits) in the offline scenario. It is formally defined as the acceptance ratio:

$$\mathcal{S} = \frac{N_{success}}{N_{total}} \tag{14}$$

where $N_{success}$ is the number of successfully embedded VNRs and $N_{total}$ is the total number of arrived requests.

- **Generalization ($\mathcal{G}$):** We define generalization conceptually as the robustness of a policy $\pi_\theta$, trained on a source topology distribution $\mathcal{D}_{\text{train}}$, when applied directly to a target distribution $\mathcal{D}_{\text{test}}$ (where $\mathcal{D}_{\text{train}} \neq \mathcal{D}_{\text{test}}$) without parameter updates. In our experiments, this is observed by evaluating the degradation or retention of performance metrics (specifically RAC, LRC, and LAR) when the agent is transferred to unseen network topologies with varying scales and structures.

- **Scalability ($\mathcal{S}\rfloor$):** We define scalability as the algorithm's capability to maintain efficient decision-making latency as the problem size increases. Formally, we analyze the growth rate of the inference time $T$ relative to the network size $|V|$ (number of nodes):

$$\mathcal{S}\rfloor \propto \frac{\partial T}{\partial |V|} \tag{15}$$

A scalable algorithm exhibits linear or near-linear growth in $T$ rather than exponential growth, ensuring feasibility in large-scale, real-world systems with high-frequency request arrivals.

Table 6: Key simulation parameters and their default values.

| Symbol | Description | Default Distribution |
|--------|-------------|----------------------|
| $\mathcal{X}_{\mathcal{N}_p}$ | Physical Node Resources | $\mathcal{U}(50, 100)$ |
| $\mathcal{X}_{\mathcal{L}_p}$ | Physical Link Bandwidth | $\mathcal{U}(50, 100)$ |
| $\mathcal{X}_{\mathcal{G}_v}$ | VN Request Size | $\mathcal{U}(2, 10)$ |
| $\mathcal{X}_{\mathcal{N}_v}$ | Virtual Node Resources | $\mathcal{U}(0, 20)$ |
| $\mathcal{X}_{\mathcal{L}_v}$ | Virtual Link Bandwidth | $\mathcal{U}(0, 50)$ |
| $\mathcal{X}_{\mathcal{I}}$ | VN Arrival Rate | $\text{Poisson}(\eta)$ |

### C.7 BENCHMARK DOCUMENTATION

To ensure usability and facilitate community adoption, we provide comprehensive and well-structured documentation website for Virne project. This documentation includes detailed guides on installation, simulation configuration, API usage, and tutorials for developing new algorithms. For the anonymous requirement of the double-blind review, we include a static version of the documentation in PDF format at https://anonymous.4open.science/r/anonymous-virne/virne-document.pdf.

## D EXPERIMENT DETAILS

### D.1 EXPERIMENTAL SETUP

We summarize the key simulation parameters in Table 6, including the settings of PN and VNs.

#### D.1.1 IMPLEMENTATION SETTINGS

We provide the details on algorithm implementation (e.g., neural networks and RL training), experimental methods on training and testing, and computing resources.

**Algorithm Implementation**. The implementation of the RL-based NFV-RA algorithm is based on PyTorch (Paszke et al., 2019). Regarding neural policies (e.g., MLP, CNN, GNN, etc), we generally set their layers to 3 and the hidden dimension of neural networks to 128. For RL optimization, we set the reward discount factor $\lambda$ to 0.99 and use the Adam optimizer with a learning rate of 0.001 and a batch size of 128. The action section during training and testing follows a sampling strategy and a greedy strategy, respectively. Additionally, we employ the $k$-shortest paths algorithm with $k = 10$ for link routing, selecting the shortest physical path that meets the bandwidth requirements for each virtual link.

**Training and Testing**. For RL-based methods, we train policies for each average arrival rate $\eta$, with random initialization of seeds in every simulation. The number of training epochs is set to 50, which is sufficient for all algorithms to converge. Typically, training a deep RL-based NFV-RA method completes within 5 hours. During testing, we evaluate the performance of all algorithms by repeating each test with 10 different random seeds (i.e., 0, 1111, 2222, ..., 9999) for each average arrival rate $\eta$, thereby ensuring statistical significance.

**Computing Resources.** We conducted all experiments on a Linux server running Ubuntu 22.04.2 LTS. This server is equipped with $8 \times$ NVIDIA H20 GPU, $128 \times$ AMD EPYC 9K84 96-Core Processor, and 1.2 TB of memory.

#### D.1.2 EVALUATED PHYSICAL TOPOLOGIES

To comprehensively evaluate the performance of NFV-RA algorithms, we conducted experiments using both simulated and real-world network topologies from SNDlib[6], widely adopted in existing studies. These topologies cover a wide range of network scales and densities, ensuring that the

---

[6]SNDlib is a well-known open-source library for telecommunication network design that offers a collection of realistic network system topologies. We use network topologies like GEANT and BRAIN from SNDlib. You can access this resource at https://sndlib.put.poznan.pl, though the specific licensing terms aren't clearly stated.

Table 7: Key Information on Evaluated Network Topologies

| Network | Node Count | Link Count | Network Density |
|---------|-----------|-----------|-----------------|
| WX100   | 100       | 500       | 0.05            |
| WX500   | 500       | 13,000    | 0.1042          |
| GEANT   | 40        | 64        | 0.0821          |
| BRAIN   | 161       | 166       | 0.0129          |

evaluation reflects diverse operational scenarios. Below, we describe the characteristics of the evaluated PN topologies.

**WX100** is a medium-scale network generated by the Waxman method (Waxman, 1988). It consists of 100 nodes connected by approximately 500 links, resulting in a density of 0.05.

**WX500** is a larger variant of the Waxman topology, extending the scale to simulate large-scale network systems. With 500 nodes and around 1300 links, it exhibits a higher density of 0.1042.

**GEANT** is a well-known academic research network, designed for high-speed data transfer across Europe. It consists of 40 nodes interconnected by 64 edges, a density of 0.0821.

**BRAIN** is a high-speed data network for scientific and cultural institutions in Berlin. It consists of 161 nodes and 166 edges, with a density of 0.0129, which is the largest topology in SDNLib.

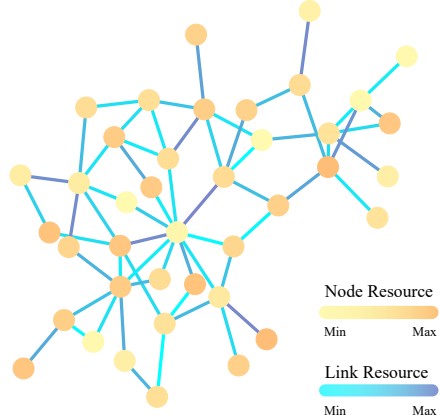

Figure 9: GEANT topology visualization.

We summarize the key characteristics of these topologies in Table 7. Furthermore, we provide a visualization of the topology of GEANT in Figure 9 to better understand the topological information.

### D.2 GENERALIZATION ON NETWORK CONDITIONS

Network conditions in real-world deployments are characterized by their inherent complexity and continuous evolution, including fluctuations in request frequencies and varying resource demands. Therefore, evaluating the generalization capabilities of trained NFV-RA policies is critical to ensure their adaptability and robust performance in diverse and dynamic network environments. Virne facilitates this by allowing pre-trained models to be tested under conditions different from their initial training setup. In this section, we investigate the generalization of various NFV-RA algorithms to two key aspects, i.e., varying traffic rates and fluctuating demand distributions. The algorithms are pre-trained under the default settings described in the main paper (Section 4.1 on Experimental Setup), and then evaluated under these new conditions. In the following experiments, to ensure figure clarity, we selected NFV-RA methods that demonstrated strong performance across various NFV-RA types in the main paper (Section 4.2.2 on Effectiveness in Online Environments).

### D.2.1 EVALUATION ON VARYING TRAFFIC RATES

To assess how well different NFV-RA algorithms adapt to changes in network load, we evaluate their performance under a range of VN request arrival rates ($\eta$). The pre-trained policies, originally trained with specific $\eta$ values for each topology (WX100 with $\eta = 0.16$, GEANT with $\eta = 0.016$, BRAIN with $\eta = 0.004$), are tested across a spectrum of $\eta$ values. For WX100, this range is from 0.04 to 0.28; for GEANT, from 0.004 to 0.028; and for BRAIN, from 0.001 to 0.007. This setup simulates scenarios from low to high network congestion, while ensuring the comparability of results. The results across WX100, GEANT, and BRAIN topologies are presented in Figure 10.

As illustrated in Figure 10 (a), (d), and (g), there is a general downward trend in the RAC for all algorithms as the average arrival rate $\eta$ increases across all three topologies. This is an expected

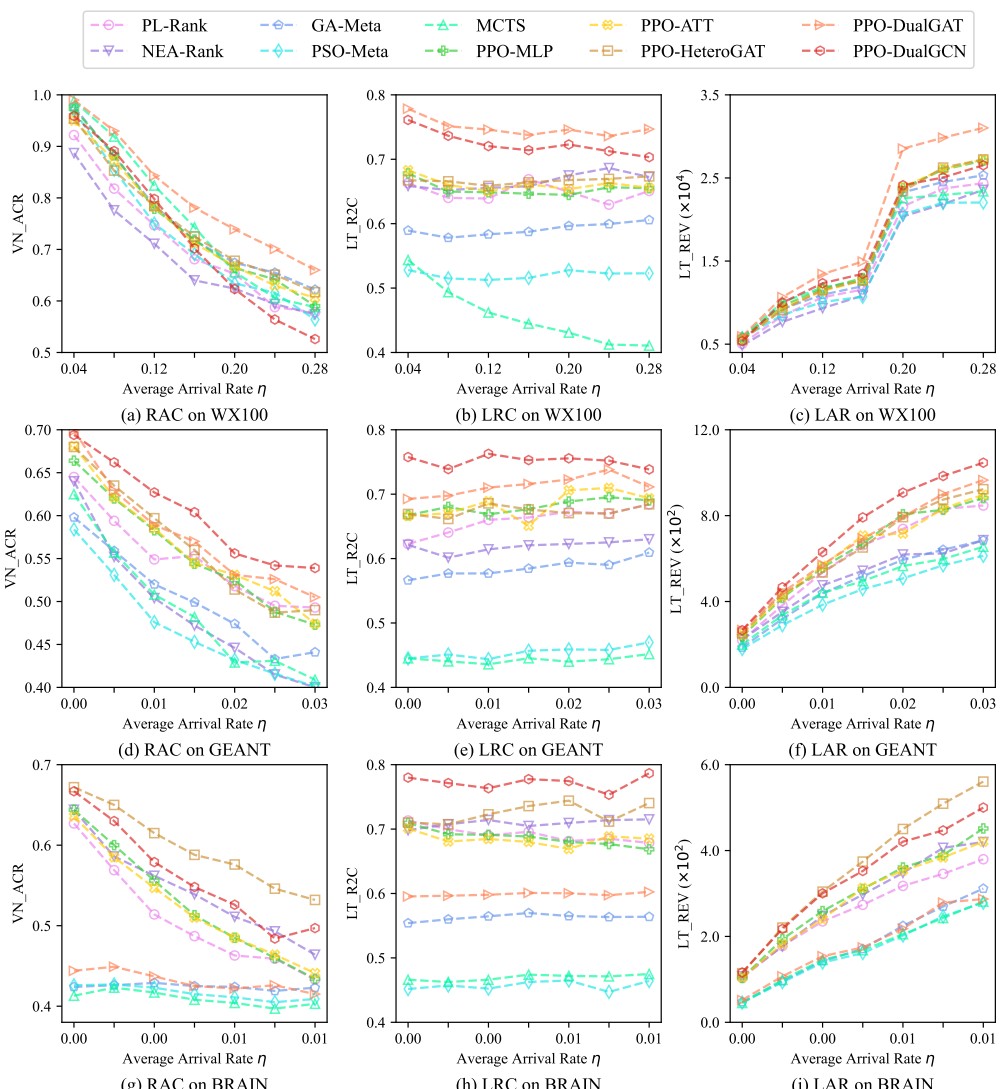

Figure 10: Results on the generalization study on varying traffic rates.

outcome, as higher traffic intensity leads to increased competition for finite physical network resources, inevitably resulting in more VN request rejections. As shown in Figure 10 (b), (e), and (h), many RL-based algorithms, particularly PPO-DualGAT and PPO-DualGCN, demonstrate remarkable stability in LRC across varying traffic rates. The results on LAR are depicted in Figure 10 (c), (f), and (I). Generally increasing with $\eta$, most algorithms process more VN requests. Algorithms like PPO-DualGAT and PPO-DualGCN consistently achieve higher LAR across the range of $\eta$ values, showcasing their superior capability in maximizing revenue even as conditions change. These experiments highlight that ❿ *while some algorithms (often sophisticated RL policies like PPO-DualGAT) exhibit graceful performance degradation and maintain high efficiency, others are more sensitive to load changes.* Additionally, from the perspective of network density, it is notable that ⓫ *the performance difference is more significant in sparser topologies, such as BRAIN, mainly due to limited feasible action space.* The denser topologies, like WX100, often have more path diversity for routing virtual links. This richer connectivity makes the embedding problem slightly less constrained, and most algorithms, including heuristics like PL-Rank, perform relatively well. However, in sparser PNs, each placement is critical because fewer alternate routes mean a poor choice can drain a key bridge's resources and trigger subsequent failures.

Overall, this study allows researchers and practitioners to select algorithms that are well-suited to the anticipated traffic dynamics of their target environments. For instance, for networks expecting highly variable loads, algorithms demonstrating flatter RAC and LRC curves across $\eta$ would be preferable.

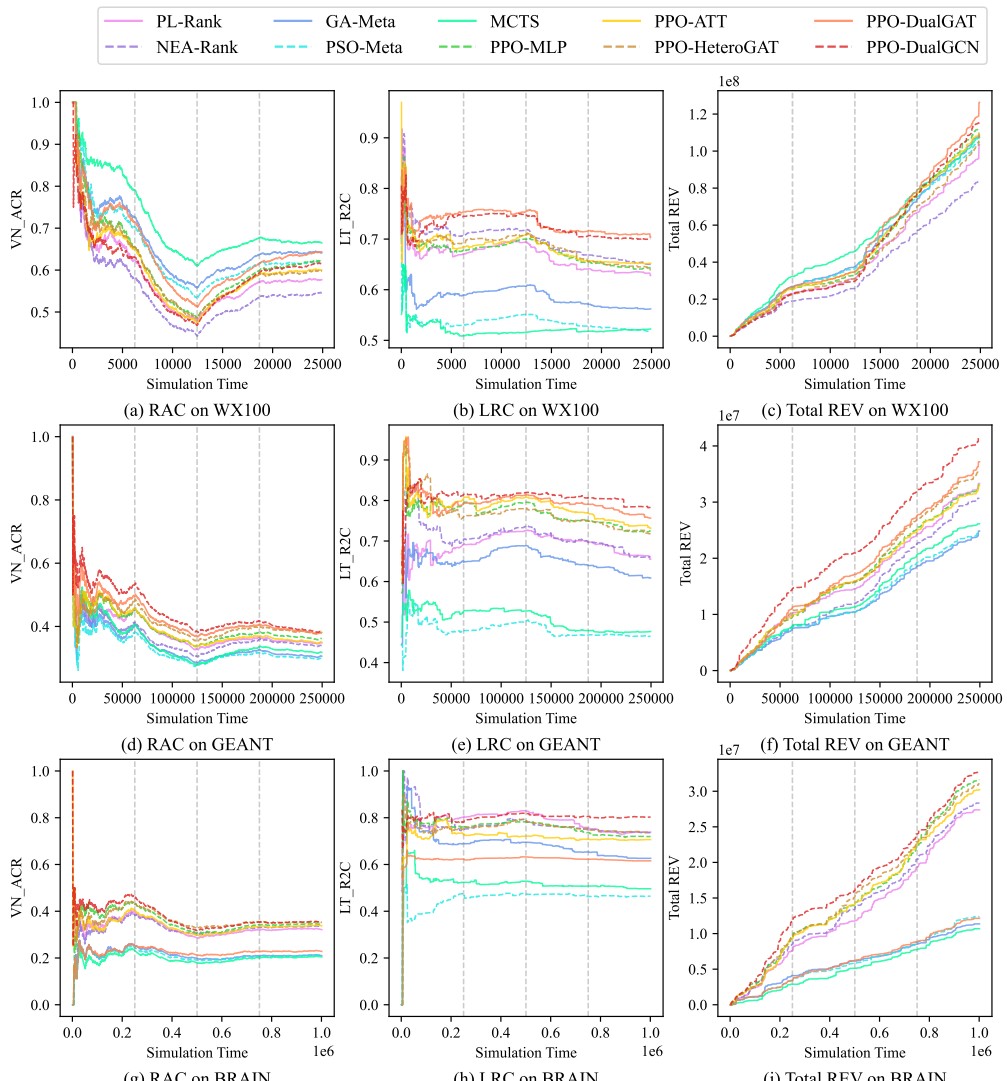

Figure 11: Results on the generalization study on fluctuating demand distribution.

### D.2.2 EVALUATION ON FLUCTUATING DEMAND DISTRIBUTION

In practical NFV environments, the characteristics of VN requests, such as the number of virtual nodes (VN size) or their resource requirements, can change over time due to evolving service demands. To simulate such dynamic conditions and evaluate algorithmic adaptability, we subject pre-trained policies to a sequence of VN requests where the underlying distribution of these requests' characteristics changes. Figure 11 illustrates the performance trends during a simulation run with these changing demands. The simulation comprises 1000 VN requests, divided into four distinct sub-groups (250 requests each). Compared to the default simulation settings, we modified one parameter related to the distribution of VN resource demand or VN node size for each subgroup: For the first group, the VN node and link resource distributions were changed to [0, 30] and [0, 75], respectively. For the second group, the VN node and link resource distributions were adjusted to [0, 40] and [0, 100], respectively. For the third group, the VN node size distribution was changed to [2, 15]. For the fourth group, the VN node size distribution was altered to [2, 20].

As illustrated in Figure 11 (a), (d), and (g) for RAC, most algorithms exhibit an initial sharp decrease, particularly as the simulation transitions through phases with increasing VN resource demands. This phase quickly consumes available physical resources, leading to lower acceptance rates. As the simulation progresses into phases where VN sizes increase, the RAC for many algorithms continues to decline or stabilizes at a lower level. This is because larger and more complex VNs are inherently

Table 8: Results on the large-scale network, WX500.

|  | RAC↑ | LRC↑ | LAR↑ | AST↓ |
|---|---|---|---|---|
| PPO-MLP | 69.50 | 0.646 | **924217.96** | 4.98 |
| PPO-CNN | 69.70 | 0.654 | 826532.68 | 5.16 |
| PPO-ATT | 68.10 | 0.647 | 923953.53 | 5.64 |
| PPO-GCN | 59.50 | 0.610 | 660552.71 | 3.80 |
| PPO-GAT | 68.80 | 0.698 | 920725.67 | 5.04 |
| PPO-GCN&S2S | 61.70 | 0.541 | 682516.68 | 4.89 |
| PPO-GAT&S2S | 58.10 | 0.540 | 699216.61 | 4.74 |
| PPO-DualGCN | **69.90** | 0.682 | 909759.61 | 5.10 |
| PPO-DualGAT | 69.80 | **0.715** | 917515.34 | 5.08 |
| PPO-HeteroGAT | 61.20 | 0.546 | 774076.11 | 7.42 |
| MCTS | 62.19 | 0.543 | 77138.32 | 18.13 |
| SA-Meta | 61.40 | 0.591 | 691030.70 | 15.09 |
| GA-Meta | 64.10 | 0.568 | 72314.48 | 16.49 |
| PSO-Meta | 68.90 | 0.566 | 750871.95 | 6.31 |
| TS-Meta | 62.10 | 0.614 | 678597.92 | 14.34 |
| NRM-Rank | 55.10 | 0.553 | 597325.51 | **0.22** |
| RW-Rank | 55.50 | 0.561 | 592502.79 | 0.24 |
| GRC-Rank | 58.50 | 0.559 | 656379.96 | **0.22** |
| PL-Rank | 64.17 | 0.632 | 79374.48 | 18.83 |
| NEA-Rank | 66.70 | 0.641 | 837297.93 | 12.97 |
| RW-BFS | 5.30 | 0.592 | 38341.41 | 0.27 |

more challenging to embed. The results on R2C shown in Figure 11 (b), (e), and (h) show a dip when the demand characteristics change, but PPO-DualGCN keeps a relatively higher LRC. For WX100 ($\eta = 0.04$) under changeable demands, PPO-DualGAT records an LRC of 0.703, whereas PPO-MLP has an LRC of 0.643. This suggests they are better at finding resource-efficient embeddings even when faced with unfamiliar or more challenging VN request types. The total revenue (i.e., cumulative LAR), depicted in Figure 11 (c), (f), and (i), shows a clear differentiation in the algorithms' ability to accumulate revenue under these dynamic conditions. Algorithms that adapt well and maintain higher RAC and LRC will naturally accrue more total revenue. Notably, advanced RL methods like PPO-DualGAT and PPO-DualGCN tend to maintain a higher performance compared to others throughout these fluctuations, indicating better adaptation. From the perspective of stage evolution, in stages 1 and 2 with increased resource demands, **⑫** *the superior performance of dual-GNN models here suggests that their learned policies latently identify action space with sufficient and rich resources within the PN to leave sufficient resources for future.* Regarding stages 3 and 4 related to increased topological complexity, this mainly estimates the ability to solve a more complex graph mapping problem. The great performance of GNN-based methods further demonstrates the importance of the representation power of neural network architectures.

Overall, these results highlight that **⑬** *policies trained on a specific VN distribution may not generalize well to others, underscoring the need to generalize not only across load levels but also across demand types.* For practical systems where service characteristics can evolve, choosing algorithms that demonstrate robustness in such fluctuating scenarios, as identifiable through Virne, is important.

## D.3 SCALABILITY ON NETWORK SIZES

The ability of an NFV-RA algorithm to scale effectively with increasing network size and complexity is paramount for its practical deployment in real-world, large-scale infrastructures. Virne facilitates a thorough assessment of scalability from two primary perspectives, i.e., (1) performance quality on large-scale network topologies, and (2) the growth trend of average solving time as both PN and VN sizes increase.

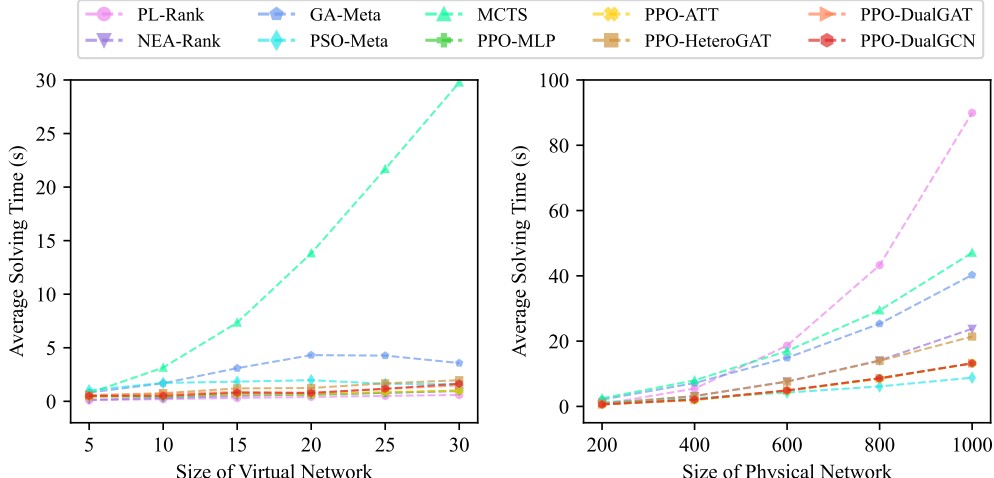

Figure 12: Results on Inference Time Scale with both VN and PN.

### D.3.1 PERFORMANCE ON LARGE-SCALE NETWORKS

To evaluate how algorithms perform when the underlying physical infrastructure is extensive, we utilize the WX500 topology within Virne. WX500, with 500 nodes and approximately 13,000 links, represents a significantly larger and denser environment compared to WX100 (100 nodes, 500 links) or other real-world topologies like GEANT (40 nodes) and BRAIN (161 nodes) used in earlier evaluations. The VN arrival rate ($\eta$) for WX500 is appropriately adjusted to reflect its increased capacity, ensuring a comparable level of resource contention. The performance of various algorithms on WX500 is detailed in Table 8.

We observe that PPO-DualGAT and PPO-DualGCN continue to exhibit strong performance on the large-scale WX500 network. This demonstrates the good scalability of their learned policies to more complex physical infrastructures. While other RL methods with simpler extractors, like PPO-MLP, exhibit limited performance, this suggests that ⓮ *the capability to effectively capture and process complex topological information becomes increasingly critical in larger networks.* Additionally, while MCTS and meta-heuristics (GA-Meta, PSO-Meta, SA-Meta, TS-Meta) can sometimes achieve competitive solution quality, their AST becomes a significant bottleneck on larger networks. Furthermore, it is interesting that ⓯ *there is also a trade-off between maximizing immediate revenue and ensuring long-term network efficiency.* Simpler models like PPO-MLP exemplify the first approach, achieving the highest total revenue (LAR) by aiming to maximize acceptance volume. In contrast, sophisticated models like PPO-DualGAT exemplify the second, securing the highest resource efficiency (LRC) with a strategic policy that prioritizes high-quality placements to preserve resource utilization efficiency.

### D.3.2 ANALYSIS OF SOLVING TIME SCALE

Beyond solution quality, the computational efficiency, measured by the Average Solving Time (AST), is a critical factor for scalability. Virne allows for detailed profiling of how AST evolves with increasing problem scales. We investigate this from two angles:

- The impact of increasing VN size (number of virtual nodes from 5 to 30) on AST, typically keeping the PN fixed (i.e., WX100).
- The impact of increasing PN size (number of physical nodes from 200 to 1000) on AST, for a consistent distribution of VN requests.

Figure 12 illustrates these relationships for various algorithms. We observe that rRL-based methods generally maintain a much flatter and lower AST curve, while meta-heuristics exhibit a substantial increase. Their inference time is primarily dictated by the forward pass through the trained neural network, which scales less dramatically with VN size compared to exhaustive search or numerous iterations. For other heuristics, they are typically the fastest, exhibiting very low and stable ASTs irrespective of VN size. While MCTS and certain meta-heuristics might achieve competitive solution

Table 9: Results on emerging networks with heterogeneous resources and latency-awareness.

| | Heterogenous Resourcing WX100 ($\eta = 0.16$) | | | | Latency-aware Edge WX100 ($\eta = 0.08$) | | | |
|---|---|---|---|---|---|---|---|---|
| | RAC↑ | LRC↑ | LAR↑ | AST↓ | RAC↑ | LRC↑ | LAR↑ | AST↓ |
| PPO-MLP | 71.10 | 0.665 | 12694.51 | 0.14 | 56.70 | 0.609 | 9792.61 | 0.20 |
| PPO-ATT | 69.30 | 0.655 | 12477.79 | 0.16 | 57.70 | 0.605 | 9460.70 | 0.21 |
| PPO-GCN | 56.30 | 0.623 | 10019.26 | 0.15 | 54.50 | 0.653 | 9045.22 | 0.20 |
| PPO-GAT | 73.40 | 0.683 | 13214.29 | 0.16 | 53.30 | 0.635 | 9570.29 | 0.26 |
| PPO-GCN&S2S | 54.20 | 0.619 | 9770.58 | 0.16 | 52.10 | 0.652 | 9144.30 | 0.22 |
| PPO-GAT&S2S | 75.00 | 0.688 | 13857.09 | 0.18 | 63.30 | 0.663 | 10703.74 | 0.22 |
| PPO-DualGCN | 66.10 | 0.715 | 12823.80 | 0.24 | 66.70 | 0.690 | 11301.84 | 0.33 |
| PPO-DualGAT | **75.10** | **0.735** | **14172.26** | 0.21 | **68.20** | **0.714** | **12056.58** | 0.32 |
| PPO-HeteroGAT | 71.20 | 0.663 | 13225.30 | 0.39 | 64.70 | 0.673 | 11563.99 | 0.40 |
| MCTS | 73.60 | 0.448 | 12847.17 | 1.72 | 52.70 | 0.480 | 8920.53 | 2.42 |
| SA-Meta | 62.10 | 0.617 | 10397.84 | 0.72 | 44.80 | 0.557 | 7789.51 | 0.80 |
| GA-Meta | 72.30 | 0.589 | 12245.59 | 1.16 | 56.10 | 0.542 | 9193.42 | 1.60 |
| PSO-Meta | 67.80 | 0.519 | 10837.89 | 1.37 | 52.50 | 0.488 | 8371.48 | 1.46 |
| TS-Meta | 65.50 | 0.650 | 11574.57 | 0.68 | 45.90 | 0.579 | 7970.68 | 0.77 |
| NRM-Rank | 60.30 | 0.525 | 9713.59 | 0.03 | 46.30 | 0.528 | 7385.33 | 0.05 |
| RW-Rank | 60.20 | 0.543 | 8947.27 | 0.02 | 39.40 | 0.554 | 7042.06 | 0.05 |
| GRC-Rank | 57.50 | 0.550 | 9008.80 | **0.01** | 42.50 | 0.556 | 7313.99 | **0.04** |
| PL-Rank | 69.70 | 0.659 | 12485.17 | 0.14 | 51.60 | 0.617 | 8546.75 | 0.23 |
| NEA-Rank | 64.60 | 0.671 | 10304.28 | 0.23 | 45.80 | 0.624 | 7367.66 | 0.34 |
| RW-BFS | 33.70 | 0.572 | 6472.31 | 0.02 | 36.70 | 0.594 | 5944.21 | 0.05 |

quality in some scenarios, their substantial computational overhead, especially with increasing problem dimensions, limits their applicability in dynamic, large-scale NFV environments. This further highlights that ⑯ *as the problem space grows exponentially, explicit search strategies at decision time become computationally prohibitive, rendering them infeasible for dynamic environments that require near-instantaneous decisions.* Additionally, ⑰ *RL-based methods, particularly those utilizing GNNs, strike a more favorable balance.* Simpler RL methods and traditional heuristics offer the highest computational speed but may compromise on optimality.

Overall, through systematic AST profiling analysis, Virne starkly highlights the trade-off between solution quality and computational scalability.

### D.4 VALIDATION ON EMERGING NETWORK

NFV is a cornerstone technology for a variety of modern and emerging network paradigms, each presenting unique characteristics and operational requirements. The Virne benchmark is designed to validate the adaptability and effectiveness of NFV-RA algorithms in such specialized environments. This section focuses on two prominent emerging scenarios, i.e., (1) networks with heterogeneous computing resources, and (2) latency-aware edge networks where delay constraints are critical. Performance in these scenarios is detailed in Table 9.

#### D.4.1 HETEROGENEOUS RESOURCING NETWORKS

Modern data centers and network infrastructures often deploy servers with diverse computing capabilities (e.g., varying CPU cores, GPU availability, memory capacities). Handling such resource heterogeneity is crucial for efficient VNF placement. Virne simulates this by configuring both PN nodes and VN nodes with multiple, distinct types of computing resources. For this evaluation, the WX100 topology is used with a VN arrival rate $\eta = 0.16$. An embedding is successful only if a physical node can satisfy all specified resource demands (i.e., CPU, GPU, and memory) of a virtual node simultaneously, adding a layer of complexity to the node mapping process.

As illustrated in Table 9, we observe that PPO-DualGAT and PPO-GAT&S2S demonstrate superior performance in heterogeneous resourcing environments, outperforming PPO-DualGCN. In heterogeneous resource scenarios, GAT-based models (e.g., PPO-DualGAT and PPO-GAT&S2S) excel because they can dynamically weigh the importance of different resource types, such as CPU versus a scarcer GPU, which is a more nuanced approach than the uniform feature treatment in GCNs.

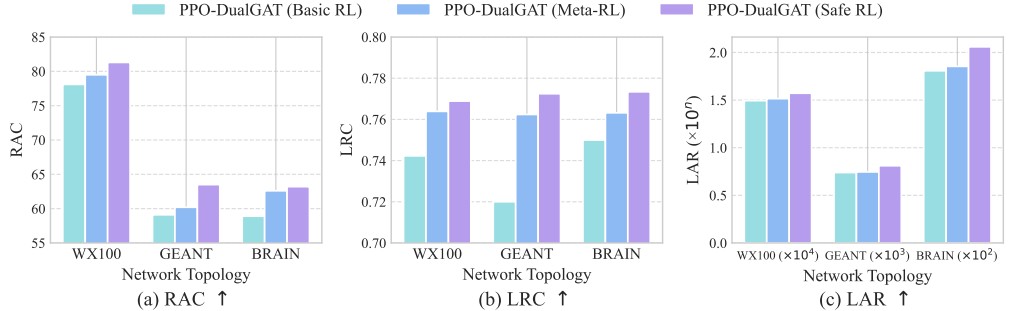

Figure 13: Performance comparison of PPO-DualGAT and its advanced variations that are augmented by the safe RL method proposed in Wang et al. (2025b) and the meta-RL proposed in Wang et al. (2024c) training methods. Please see their MDP formulation in Appendix C.2. The results show that both the safe RL and meta-RL variants consistently outperform the basic algorithm, with the safe RL approach achieving higher improvements. This underscores the importance of robust constraint management and generalization in solving complex resource allocation problems, which are highlighted in this section for future algorithmic innovation.

This highlights that ❶❽ *attention mechanisms over features and neighbors are highly effective for multi-dimensional node-level constraints*. Simpler RL architectures like PPO-MLP and PPO-GCN, while competent, tend to show a slight dip in performance compared to scenarios with homogeneous resources. This reveals that ❶❾ *Effectively managing and balancing multiple, diverse resource requirements simultaneously poses a greater challenge for architectures that may not explicitly differentiate or dynamically weigh these varied resource dimensions.*

### D.4.2 Latency-aware Edge Networks

Edge computing networks bring computational resources closer to end-users, making them ideal for latency-sensitive applications. In such scenarios, NFV-RA algorithms must not only allocate resources efficiently but also strictly adhere to the delay requirements of services. This evaluation uses the WX100 topology with a VN arrival rate $\eta = 0.08$, which could represent a typical edge deployment with specific traffic characteristics. The critical factor introduced here is a latency constraint for each virtual link. When a virtual link is mapped to a physical path, the cumulative propagation delay of the physical links constituting that path must not exceed the virtual link's specified maximum tolerable latency, i.e., 100ms. This necessitates that the algorithms should possess the ability to manage constraint management. We additionally consider the latency attributes as the input features of neural networks for all deep RL-based methods.

As illustrated in Table 9, the introduction of latency constraints generally leads to lower RAC and LRC values across all algorithms compared to scenarios without such strict path requirements. However, PPO-DualGAT again shows the most robust performance, achieving the highest RAC (68.20%), LRC (0.714), and LAR (12056.58). In contrast, traditional heuristics and MCTS struggle to satisfy resource and latency simultaneously, suffering larger drops. This reveals that ❷⓪ *The fundamental weakness of traditional heuristics is exposed in scenarios with path-level constraints.* Their decoupled "rank nodes, then find paths" approach is brittle, as an optimal node choice based on local metrics can make it impossible to subsequently find a valid low-latency path, leading to cascading failures.

## E Discussion on Future Directions

The empirical analysis enabled by the Virne benchmark illuminates several promising avenues for advancing deep RL-based NFV-RA methods. Although state-of-the-art algorithms show substantial capability, significant challenges remain that hinder more robust, efficient, and practical solutions. In this section, we outline several key future directions, and we provide an empirical study of an emerging algorithm addressing some of these directions in Figure 13.

### E.1 REPRESENTATION LEARNING FOR CROSS-GRAPH STATUS

Future research should focus on developing more sophisticated representation learning techniques to capture the intricate, dynamic interplay between VN requirements and PN states, including the evolving mapping status itself. While Virne demonstrates the strength of dual-GNN architectures in processing VN and PN features separately, there is significant potential in exploring emerging architectures such as Transformers and diffusion models. These advanced models could enable the learning of even richer cross-graph relational embeddings and path-level attribute awareness (Wang et al., 2019; Guo et al., 2025). Specifically, they may offer superior capabilities in capturing long-range dependencies across multiple resource types and handling complex constraints, potentially leading to more nuanced and context-aware embedding decisions. Techniques that explicitly model the partially embedded state of VNs and the resultant constraints on future placements are crucial, and Virne's extensible architecture provides a ready platform for benchmarking these novel designs.

### E.2 ROBUST LEARNING FOR COMPLEX AND HARD CONSTRAINTS

The management of multifaceted and often conflicting constraints (e.g., resource capacity, latency, reliability, energy consumption) remains a central challenge in NFV-RA, which requires guaranteeing zero-violation. Thus, it is significant to explore constraint-aware RL frameworks that more explicitly model and navigate these hard constraints. This could involve advancing constrained policy optimization methods and developing novel reward structures that better reflect constraint satisfaction (Gu et al., 2022; Wang et al., 2025b). The goal is to train safe policies that are not only effective in optimizing primary objectives but are also inherently robust in satisfying diverse operational constraints.

### E.3 SCALABILITY IN EXTREMELY LARGE-SCALE NETWORKS

Virne's scalability studies demonstrate that while current RL methods, particularly GNN-based ones, scale better than many traditional approaches, their computational and memory demands can still grow significantly with network size. Addressing NFV-RA in truly massive, carrier-grade networks requires further breakthroughs in algorithmic scalability. Future directions could include exploring hierarchical RL where policies operate at different levels of abstraction, building a non-autoregressive solution modeling method (Bengio et al., 2021), or designing GNN architectures that are less sensitive to the global network scale (Wang et al., 2024a). Methods that can learn transferable knowledge or localizable policies that effectively stitch together solutions for very large infrastructures are highly desirable (Geng et al., 2023).

### E.4 GENERALIZABLE POLICY CROSS VARYING SCENARIOS

Achieving policies that generalize effectively not only to unseen network topologies but also across varying scales (both PN and VN sizes) and highly dynamic operational conditions (e.g., non-stationary demand patterns, unexpected resource outages) is a critical frontier. To facilitate research in this direction, Virne explicitly incorporates a Multi-Task MDP framework (see Appendix C.2.1), allowing users to define diverse tasks encompassing different physical topologies, resource distributions, and traffic patterns. As demonstrated by the meta-learning results in Figure 13, this foundation enables the training of agents capable of adapting to new environments. Future research should leverage this flexible task definition to further investigate advanced techniques such as meta-RL (Beck et al., 2023), curriculum learning to incrementally expose agents to more complex scenarios, and domain randomization to enhance robustness against a wider array of unseen conditions (Zhou et al., 2023; Wang et al., 2024c). The ultimate aim is to develop omni-generalizable NFV-RA solutions that require minimal retraining for deployment in diverse and evolving network environments.

## F LIMITATIONS ON SIM-TO-REAL GAP

As a simulation-based benchmark, Virne makes specific abstractions to ensure the computational efficiency required for training Deep RL agents. There are several primary gaps between our default simulation and physical deployment:

- **Simulation Granularity (Flow vs. Packet).** Virne functions as a flow-level orchestration simulator. To maintain high training throughput, it abstracts packet-level dynamics (e.g., microsecond queuing delays, jitter) into statistical link attributes. While this simplifies latency modeling, it allows for rapid algorithmic iteration.

- **Infrastructure Observability (Global vs. Partial)**. The default experimental settings assume a centralized SDN controller with global state visibility. In contrast, real-world large-scale networks may only offer partial observability. Virne mitigates this via a modular architecture, allowing users to apply feature masking to simulate POMDP scenarios without altering the core engine.

- **Resource Dynamics (Idealized vs. Noisy)**. The default environment assumes deterministic resource allocation. Physical systems, however, often experience fluctuations due to background processes or hardware overhead ("noisy neighbors"). Virne supports extensibility in its transition function, allowing users to inject stochastic noise to model these environmental perturbations.

## G   THE USE OF LARGE LANGUAGE MODELS STATEMENT

The authors use Large Language Models (LLMs) as an assistive tool in the preparation of this manuscript, in accordance with the ICLR 2026 policy. The core roles of LLMs are described as follows. First, we utilize the LLM-powered search and summarization tools to survey related works and understand the current state of the field. Second, we use LLM-based coding assistants to assist in coding and suggest optimizations, thereby accelerating the development of the Virne codebase. Third, while preparing the manuscript, we use LLMs to proofread, check grammar, and refine the language in the manuscript for improved clarity and readability.

