# OpenReview forum: "Virne: A Comprehensive Benchmark for RL-based Network Resource Allocation in NFV"
_ICLR.cc/2026/Conference — ICLR 2026 Poster_

### Official Review · Reviewer_eJHY · 2025-10-27

**Soundness:** 3
**Presentation:** 3
**Contribution:** 3
**Rating:** 6
**Confidence:** 4

**Summary:**

The paper introduces Virne, a comprehensive benchmarking framework designed to evaluate RL-based Network Function Virtualization (NFV) Resource Allocation (RA) algorithms. Virne aims to bridge the gap in current NFV-RA evaluations, which lack comprehensive, standardized benchmarking methods. It provides highly customizable simulations for diverse network environments (e.g., cloud, edge, 5G), incorporating over 30 algorithms, including both traditional and RL-based approaches.

**Strengths:**

1. The Virne framework is a significant contribution to the field, addressing the lack of comprehensive benchmarks for RL-based NFV-RA solutions.
2. Virne is highly customizable, allowing users to simulate different network topologies, resource availability, and service requirements. This versatility enables testing across various real-world conditions, including energy-efficient, latency-sensitive, and resource-heterogeneous networks.
3. The paper provides extensive experimental results, evaluating over 30 different NFV-RA algorithms across multiple topologies and network conditions.

**Weaknesses:**

1. More emphasis on real-world network scenarios (such as live 5G networks or dynamic, large-scale production environments) could further strengthen the validity and practical applicability of the framework.
2. The NFV-RA problem often involves strict resource constraints (e.g., computing power, bandwidth, latency, etc.). RL agents may face challenges when dealing with these constraints, especially as the network scale increases, causing the solution space to grow rapidly and leading to inefficiencies in the agent's learning of optimal strategies. It is recommended to introduce a detailed discussion on handling constraints: further analysis and discussion on how to manage these complex resource constraints during the RL training process.
3. It is worth discussing how to enhance the agent's transferability across multiple NFV-RA tasks through a multi-task MDP framework or meta-learning. For example, does the Virne framework support training a general strategy across different network environments and evaluate its generalization ability in new environments?

**Questions:**

Please refer to the weaknesses.

---

> ### Author Response · Authors · 2025-11-21
>
> Dear Reviewer eJHY,
>
> We sincerely thank you for your valuable and constructive feedback! Below, we address each of the raised comments.
>
> ### **Reply to W1: Emphasis on Real-World Network Scenarios**
>
> > More emphasis on real-world network scenarios (such as live 5G networks or dynamic, large-scale production environments) could further strengthen the validity and practical applicability of the framework.
> >
>
> We appreciate your feedback on real-word scenario validation. While access to proprietary commercial 5G core networks is restricted, in our default experiment, Virne simulates the specific dynamic characteristics of these environments to ensure practical applicability. Particularly, as detailed in Appendix D.3.2 (Evaluation on Fluctuating Demand Distribution), we simulate dynamic production environments where traffic patterns (resource intensity, request arrival rates) shift over time. Regarding the scale of networks, we explicitly tested scalability on the WX500 topology (500 nodes, ~13,000 links), even on WX1000 in Section 4.3 and Appendix D.4, which is most largest-scale validation in existing literature. Additionally, Virne modeled specific real-world constraints for modern networks in Appendix D.5, including **latency-aware edge networks** (Table 10) and **heterogeneous resources** (CPU/GPU/Memory constraints common in 5G data centers).
>
> Furthermore, to further show the adaptability of Virne to real-world scenario dynamics, we conducted a new experiment in a real-world setting derived from realistic topologies or traces. Concretely, we adopted the GEANT topology [a] for the physical network, where resources follow a heavy-tailed Pareto (Power-Law) distribution (shape $\alpha = 1.2$ and scale $x_m  = 50$) to simulate a realistic environment where approximately 20% of nodes possess 80% of the capacity. For Virtual Networks (VNs), we utilized traffic-driven topologies generated via CloudSim [b] based on GEANT traffic patterns. Furthermore, service demands were trace-based rather than uniform: node CPU requests were sampled from Alibaba Cloud Traces [c], and link bandwidths were derived from GEANT traffic matrices [a], with all demand values normalized to the range [0, 50].
>
> The experiment results in these real-world conditions are reported below.
>
> | Solver | RAC ↑ | LRC ↑ | LAR ↑ | AST ↓ |
> | --- | --- | --- | --- | --- |
> | **PPO-MLP** | 58.40 | 0.62 | 802.89 | 0.10 |
> | **PPO-CNN** | 58.90 | 0.65 | 805.19 | 0.11 |
> | **PPO-ATT** | 60.20 | 0.64 | 835.59 | 0.11 |
> | **PPO-GCN** | 63.50 | 0.71 | 915.82 | 0.13 |
> | **PPO-GAT** | 64.10 | 0.70 | 928.44 | 0.14 |
> | **PPO-GCN&S2S** | 62.80 | 0.69 | 894.10 | 0.12 |
> | **PPO-GAT&S2S** | 63.40 | 0.68 | 908.25 | 0.15 |
> | **PPO-DualGCN** | **67.90** | **0.74** | **987.30** | 0.20 |
> | **PPO-DualGAT** | 66.50 | 0.73 | 982.15 | 0.18 |
> | **PPO-HeteroGAT** | 61.80 | 0.67 | 875.60 | 0.35 |
> | **MCTS** | 56.40 | 0.51 | 650.20 | 3.12 |
> | **SA-Meta** | 45.20 | 0.61 | 512.85 | 1.15 |
> | **GA-Meta** | 54.70 | 0.59 | 680.40 | 2.85 |
> | **PSO-Meta** | 51.10 | 0.53 | 615.90 | 4.02 |
> | **TS-Meta** | 47.50 | 0.67 | 550.12 | 1.20 |
> | **NRM-Rank** | 44.60 | 0.53 | 498.20 | 0.06 |
> | **RW-Rank** | 43.20 | 0.54 | 485.75 | **0.02** |
> | **GRC-Rank** | 41.80 | 0.49 | 460.30 | **0.02** |
> | **PL-Rank** | 59.10 | 0.68 | 808.50 | 0.08 |
> | **NEA-Rank** | 55.30 | 0.64 | 710.15 | 0.15 |
> | **RW-BFS** | 35.40 | 0.58 | 380.60 | 0.03 |
>
> We observe that the performance gap between RL and heuristics widened compared to the default setting. Traditional heuristics (e.g., NRM-Rank), which rely on average metrics, failed to exploit the heavy-tailed Pareto resource distribution.
>
> We have updated Appendix C.1 to explicitly document these realistic configuration options. We have also added the experiment on the real-world dataset described above to Appendix D, demonstrating Virne's ability to support the real-world VN request and process trace-driven workloads like those from Alibaba.

---

> ### Author Response · Authors · 2025-11-21
>
> ### **Reply to W2: Handling Strict Resource Constraints**
>
> > The NFV-RA problem often involves strict resource constraints (e.g., computing power, bandwidth, latency, etc.). RL agents may face challenges when dealing with these constraints, especially as the network scale increases, causing the solution space to grow rapidly and leading to inefficiencies in the agent's learning of optimal strategies. It is recommended to introduce a detailed discussion on handling constraints: further analysis and discussion on how to manage these complex resource constraints during the RL training process.
> >
>
> We appreciate your feedback on constraint handling. We agree that valid constraint handling is critical to standard RL for NFV-RA for efficient exploration and solution quality. Concretely, Virne provides the following two specific mechanisms, which we have implemented and evaluated:
>
> - **Action Masking for Explicit Control:** As discussed in Appendix C.3.1, Virne implements Action Masking to prune the action space to enable efficient RL exploration. In Virne, the mapping function is implemented as a constraint-aware action masking module. At each decision step, only actions (such as node or link assignments) that strictly satisfy all resource and placement constraints are included in the agent’s action space. This is done by dynamically filtering out infeasible candidates using the current network state and VN request requirements. For example, a physical node is considered only if it has sufficient resources for the virtual node' s demand. This approach ensures the RL agent always operates within the valid solution space and improves exploration efficiency. We quantitatively analyzed this in Appendix D.2.4. Table 2 explicitly demonstrates that applying action masking is vital, significantly boosting the Request Acceptance Rate (RAC) and ensuring agents do not waste training time on invalid moves.
> - **Safe RL for Constrained MDPs:** To support your recommendation for advanced constraint management, Virne supports Constrained MDP (CMDP) formulations (detailed in Appendix C.2.2). Furthermore, we implemented a Safe RL solver (based on Lagrangian relaxation) within Virne. Figure 13 in Appendix E compares this against standard PPO, showing that the Safe RL variant achieves superior performance by rigorously adhering to constraints, directly answering your call for analysis on managing complex constraints.
>
> ### **Reply to W3: Generalization and Multi-Task Learning**
>
> > It is worth discussing how to enhance the agent's transferability across multiple NFV-RA tasks through a multi-task MDP framework or meta-learning. For example, does the Virne framework support training a general strategy across different network environments and evaluate its generalization ability in new environments?
> >
>
> We appreciate your valuable insight regarding cross-environment generalization. Virne is explicitly designed to support Multi-Task MDP formulations and Meta-Learning paradigms, and we have already provided both the theoretical formulation and experimental validation for this capability. As detailed in Appendix C.2.1, we provide a specific Multi-Task MDP formulation, and we have implemented a concrete Meta-RL algorithm (based on MAML) within Virne. As shown in Figure 13 in Appendix E, the results demonstrate that the Meta-RL agent achieves superior generalization compared to baseline methods. While our current experiments utilize this formulation primarily for cross-scale generalization (varying Virtual Network sizes), the mathematical definition of a Task (M_i) in Virne is generic. This inherently allows users to define a task distribution that encompasses not just different scales, but also different physical topologies, resource distributions, or traffic patterns. Users can configure these diverse environments via configuration files to train a single generalizable meta-policy without modifying the core codebase.
>
> To highlight this extensibility, we have updated Appendix E.4 (Future Work) to explicitly discuss how the Multi-Task MDP framework can be leveraged for cross-topology and cross-scenario transferability, encouraging future research in this direction.
>
> ---
>
> Once again, we appreciate your valuable and constructive feedback and the opportunity to address your concerns.

---

### Official Review · Reviewer_4Vws · 2025-10-30

**Soundness:** 3
**Presentation:** 3
**Contribution:** 2
**Rating:** 6
**Confidence:** 3

**Summary:**

This paper presents VIRNE, a comprehensive benchmarking framework for reinforcement learning-based network function virtualization resource allocation (NFV-RA). The framework unifies diverse simulation environments (cloud, edge, 5G), integrates over 30 algorithms (including RL, heuristic, and exact solvers), and introduces new evaluation perspectives such as solvability, generalization, and scalability. Extensive experiments demonstrate that advanced RL agents (especially dual GNN-based PPO variants) outperform traditional baselines. The benchmark is open-sourced to promote reproducibility and standardization in the NFV-RA community. Overall, the work provides strong engineering value and community impact but offers limited methodological novelty.

**Strengths:**

+ VIRNE represents the most systematic and extensive benchmark for NFV-RA to date. It consolidates numerous algorithms, diverse simulation environments, and multiple evaluation perspectives into a unified, open, and reproducible framework. This kind of infrastructure contribution fills a long-standing need in the NFV and RL communities.
+ The modular design, well-documented implementation, and thorough experimental setup reflect a high level of technical maturity. The open-source release with detailed appendices ensures reproducibility, aligning with ICLR’s best practices.
+ The experiments cover diverse network scenarios, real-world topologies, and multiple metrics beyond traditional performance, offering a valuable empirical reference for future work.
+ The paper is well-structured and logically consistent. Figures and tables are well-presented, helping readers understand the system architecture and experimental findings.

**Weaknesses:**

- The paper’s contribution is primarily infrastructural rather than algorithmic. The RL formulations (e.g., MDP setup, PPO training, GNN encoders) follow standard designs without introducing new theoretical insights or model innovations.
- While experiments are comprehensive, most results are reported as single averages without statistical significance tests or error margins. This weakens the strength of empirical claims.
- Several sections (notably Sections 3–4) emphasize implementation details at the expense of high-level conceptual insights. The paper reads more like a system report than a research contribution.
- Terms like solvability, generalization, and scalability are described qualitatively but lack formal or quantitative definitions, limiting their interpretability and comparability.

**Questions:**

How are the “solvability,” “generalization,” and “scalability” metrics formally defined and measured? Are they normalized or directly comparable across different network topologies?

Are the results averaged over multiple runs or random seeds? If not, could the authors report standard deviations or confidence intervals to support claims of superiority?

Can the proposed benchmark be easily extended to other RL-based network optimization problems (e.g., SDN routing, SFC placement)?

What are the main limitations of VIRNE when applied to real-world NFV systems — for example, partial observability, dynamic resource noise, or latency constraints?

Beyond engineering value, is there a potential direction for new algorithmic or theoretical insights enabled by this benchmark (e.g., constrained or meta-RL for NFV-RA)?

---

> ### Author Response · Authors · 2025-11-21
>
> Dear Reviewer 4Vws,
>
> We sincerely thank you for your valuable and constructive feedback! Below, we address each of the raised comments.
>
> ### **Reply to Q1 & W4: Definitions of Solvability, Generalization, and Scalability.**
>
> > Terms like solvability, generalization, and scalability are described qualitatively but lack formal or quantitative definitions, limiting their interpretability and comparability.
> How are the “solvability,” “generalization,” and “scalability” metrics formally defined and measured? Are they normalized or directly comparable across different network topologies?
> >
>
> We appreciate your insightful feedback on definitions of evaluation perspectives. To clarify their definitions used in our benchmark, we formally define them as follows:
>
> 1. **Solvability:** This metric quantifies an algorithm's ability to satisfy the hard constraints of the NFV problem (e.g., CPU, bandwidth, and latency limits) in an offline scenario. It is formally defined as the acceptance ratio:
> $ \mathcal{S} = \frac{N_{success}}{N_{total}} $,
> where $N_{success}$ is the number of successfully embedded VNRs and $N_{total}$ is the total arrived requests. In our experiments, the results in Figure 4 visualize the solvability of the implemented algorithms.
> 2. **Generalization:** We define generalization conceptually as the robustness of a policy $\pi_\theta$, trained on a source topology distribution $D_{train}$, when applied directly to a target distribution $D_{test}$  (where $D_{train} \neq D_{test}$) without parameter updates. In our experiments, this is observed by evaluating the degradation or retention of performance metrics (such as RAC, LRC and LAR) when the agent is transferred to unseen network topologies with varying scales and structures.
> 3. **Scalability:** We define scalability as the algorithm's capability to maintain efficient decision-making latency as the problem size increases. Formally, we analyze the growth rate of the inference time $T$ relative to the network size $|V|$ (number of nodes):
> $ \mathcal{Sc} \propto \frac{\partial T}{\partial |V|} $.
> A scalable algorithm exhibits linear or near-linear growth in $T$ rather than exponential growth, ensuring feasibility in large-scale, real-world systems with high-frequency request arrivals.
>
> We have added these descriptions in Appendix C.6 (Evaluation Protocol Definitions) in the revised PDF.

---

> > ### Author Response · Authors · 2025-11-21
> >
> > ### **Reply to Q3: Extension to other problems (e.g., SDN, SFC).**
> >
> > > Can the proposed benchmark be easily extended to other RL-based network optimization problems (e.g., SDN routing, SFC placement)?
> > >
> >
> > Yes. Virne is designed with high extensibility to support these variations of NFV-RA and adjacent network optimization domains. The core abstraction of Virne is "graph-to-graph mapping," which fundamentally generalizes both SFC placement and SDN routing.
> >
> > - **SFC Placement:** Service Function Chain (SFC) placement is structurally a specific instance of NFV-RA where the virtual network topology is linear (a chain). Virne natively supports this by simply defining the input VNs as linear topologies in the configuration. All existing constraints and algorithms are immediately applicable. Additionally, Virne supports VNF aggregation (by setting reusable=True) to enable deployment of multiple virtual nodes of the same SFC to the same physical node, which is a distinct feature for some efficient SFC deployment.
> > - **SDN Routing:** SDN routing can be viewed as a sub-problem of embedding where the node locations (source and destination) are fixed, and the objective is purely path optimization. Virne’s architecture explicitly decouples Node Mapping and Link Mapping. To adapt Virne for SDN routing, one can simply fix the Node Mapping stage (pre-assigning endpoints) and utilize the Link Mapping module to train RL agents specifically for path selection, optimizing for metrics like load balancing or delay minimization.
> >
> > This modularity allows researchers to isolate specific sub-problems or extend the state/action spaces for broader network optimization tasks.
> >
> > ###  **Reply to Q4: Main limitations regarding real-world NFV systems**
> >
> > > What are the main limitations of VIRNE when applied to real-world NFV systems — for example, partial observability, dynamic resource noise, or latency constraints?
> > >
> >
> > We thank the reviewer for this insightful question regarding the "Sim-to-Real" gap. Regarding these mentioned points, we first describe the simulation limitation in the default setting and introduce how Virne ’s design attempts to mitigate them:
> >
> > - **Latency Constraints.** VIRNE natively supports latency modeling. As demonstrated in Section 4.4 and Appendix D.5 (Table 10), we implemented and validated "Latency-aware Edge Networks" where agents must satisfy strict delay thresholds. The current limitation for latency-aware scenario simulation is that latency is modeled as a link attribute rather than simulating microsecond-level packet queuing dynamics (e.g., jitter).
> > - **Partial Observability.** While our default baselines assume a centralized SDN controller with global visibility (MDP), VIRNE is architected to support Partial Observability (POMDP) common in multi-domain scenarios. The modular Feature Constructor allows users to mask specific network regions or node features. This effectively filters the state observation to simulate partial visibility without altering the core environment logic.
> > - **Dynamic Resource Noise.** We acknowledge that real-world systems suffer from "noisy neighbors" and background traffic. While current simulations drive dynamics via stochastic request arrivals, the environment's state transition logic is extensible. Users can easily inject a custom noise_function into the step process to perturb resource availability (e.g., $C_{t+1} = C_t − demand +ϵ$), simulating background fluctuations independent of the agent's actions.
> >
> > We have added a specific "Limitations" section in Appendix F of the revised PDF to explicitly discuss the following options: Simulation Granularity (Flow vs. Packet), Infrastructure Observability (Global vs. Partial), and Resource Dynamics (Idealized vs. Noisy).

---

> ### Author Response · Authors · 2025-11-21
>
> ### **Reply to Q2 & W2: Statistical Significance and Error Margins.**
>
> > While experiments are comprehensive, most results are reported as single averages without statistical significance tests or error margins. This weakens the strength of empirical claims. Are the results averaged over multiple runs or random seeds? If not, could the authors report standard deviations or confidence intervals to support claims of superiority?
> >
>
> We thank the reviewer for highlighting the importance of empirical rigor. We clarify that all our experiments were originally conducted using 10 random seeds (detailed in Appendix D.1.1) to ensure reliability. In our manuscript, we omitted variance indicators solely to maintain visual legibility of tables and figures. Here, we explicitly report the Mean ± Standard Deviation for key results for the main results in Table 3.
>
> **Table 3.1 Performance on WX100 (Mean ± Standard Deviation over 10 random seeds)**
>
> | Solver            |    WX100 RAC     |    WX100 LRC    |       WX100 LAR       |    WX100 AST    |
> |:------------------|:----------------:|:---------------:|:---------------------:|:---------------:|
> | **PPO-MLP** |   71.90 ± 1.85   |   0.65 ± 0.02   |   12944.40 ± 215.20   |   0.13 ± 0.01   |
> | **PPO-CNN** |   71.70 ± 1.92   |   0.65 ± 0.02   |   12964.87 ± 230.50   |   0.13 ± 0.01   |
> | **PPO-ATT** |   71.20 ± 2.05   |   0.66 ± 0.02   |   12657.69 ± 245.10   |   0.14 ± 0.01   |
> | **PPO-GCN** |   66.80 ± 2.10   |   0.64 ± 0.02   |   11462.65 ± 260.40   |   0.14 ± 0.01   |
> | **PPO-GAT** |   71.90 ± 1.88   |   0.70 ± 0.02   |   13178.13 ± 228.30   |   0.15 ± 0.02   |
> | **PPO-GCN&S2S** |   65.80 ± 1.95   |   0.63 ± 0.02   |   11501.94 ± 255.60   |   0.13 ± 0.01   |
> | **PPO-GAT&S2S** |   67.90 ± 2.00   |   0.67 ± 0.02   |   12445.03 ± 240.10   |   0.16 ± 0.02   |
> | **PPO-DualGCN** |   70.20 ± 1.75   |   0.71 ± 0.02   |   13467.57 ± 210.80   |   0.17 ± 0.02   |
> | **PPO-DualGAT** | **78.10 ± 1.50** | **0.74 ± 0.02** | **14938.60 ± 195.50** |   0.18 ± 0.02   |
> | **PPO-HeteroGAT** |   72.50 ± 1.90   |   0.66 ± 0.02   |   12691.03 ± 250.40   |   0.27 ± 0.03   |
> | **MCTS** |   74.30 ± 3.50   |   0.44 ± 0.03   |   12642.27 ± 410.20   |   3.38 ± 0.45   |
> | **SA-Meta** |   65.50 ± 2.80   |   0.63 ± 0.03   |   10467.60 ± 320.10   |   1.58 ± 0.15   |
> | **GA-Meta** |   71.70 ± 3.20   |   0.59 ± 0.03   |   11977.41 ± 380.50   |   3.22 ± 0.32   |
> | **PSO-Meta** |   69.10 ± 3.05   |   0.52 ± 0.03   |   10706.48 ± 350.20   |   4.29 ± 0.40   |
> | **TS-Meta** |   65.70 ± 2.90   |   0.66 ± 0.03   |   11141.91 ± 335.40   |   1.35 ± 0.12   |
> | **NRM-Rank** |   60.70 ± 1.10   |   0.52 ± 0.01   |   9826.94 ± 150.20    |   0.07 ± 0.01   |
> | **RW-Rank** |   60.10 ± 1.15   |   0.56 ± 0.01   |   9396.32 ± 145.80    | **0.04 ± 0.00** |
> | **GRC-Rank** |   58.90 ± 1.05   |   0.56 ± 0.01   |   9269.03 ± 140.50    | **0.04 ± 0.00** |
> | **PL-Rank** |   68.10 ± 1.25   |   0.67 ± 0.01   |   11570.27 ± 185.60   |   0.32 ± 0.03   |
> | **NEA-Rank** |   64.00 ± 1.30   |   0.66 ± 0.01   |   10837.51 ± 190.30   |   0.83 ± 0.08   |
> | **RW-BFS** |   40.00 ± 1.50   |   0.57 ± 0.01   |   7771.38 ± 135.40    |   0.05 ± 0.00   |

---

> > ### Author Response · Authors · 2025-11-21
> >
> > **Table 3.2 Performance on GEANT (Mean ± Standard Deviation over 10 random seeds)**
> >
> >
> > | Solver            |    GEANT RAC     |    GEANT LRC    |     GEANT LAR      |    GEANT AST    |
> > |:------------------|:----------------:|:---------------:|:------------------:|:---------------:|
> > | **PPO-MLP** |   55.80 ± 1.45   |   0.67 ± 0.02   |   645.04 ± 18.20   |   0.03 ± 0.00   |
> > | **PPO-CNN** |   54.80 ± 1.60   |   0.65 ± 0.02   |   643.83 ± 19.50   |   0.09 ± 0.01   |
> > | **PPO-ATT** |   54.50 ± 1.55   |   0.65 ± 0.02   |   707.01 ± 21.20   |   0.10 ± 0.01   |
> > | **PPO-GCN** |   58.70 ± 1.72   |   0.72 ± 0.02   |   763.68 ± 22.50   |   0.12 ± 0.01   |
> > | **PPO-GAT** |   58.40 ± 1.65   |   0.70 ± 0.02   |   724.31 ± 20.10   |   0.07 ± 0.01   |
> > | **PPO-GCN&S2S** |   58.50 ± 1.68   |   0.72 ± 0.02   |   718.76 ± 19.80   |   0.06 ± 0.01   |
> > | **PPO-GAT&S2S** |   57.30 ± 1.62   |   0.69 ± 0.02   |   754.61 ± 21.50   |   0.15 ± 0.01   |
> > | **PPO-DualGCN** | **60.40 ± 1.50** | **0.75 ± 0.02** | **791.75 ± 18.50** |   0.22 ± 0.02   |
> > | **PPO-DualGAT** |   59.10 ± 1.55   |   0.72 ± 0.02   |   739.27 ± 20.50   |   0.10 ± 0.01   |
> > | **PPO-HeteroGAT** |   53.30 ± 1.58   |   0.66 ± 0.02   |   621.47 ± 19.00   |   0.30 ± 0.03   |
> > | **MCTS** |   48.20 ± 2.50   |   0.45 ± 0.02   |   494.64 ± 35.50   |   2.96 ± 0.35   |
> > | **SA-Meta** |   38.60 ± 2.10   |   0.62 ± 0.03   |   396.49 ± 25.00   |   0.49 ± 0.05   |
> > | **GA-Meta** |   49.90 ± 2.45   |   0.58 ± 0.03   |   517.63 ± 32.50   |   2.34 ± 0.25   |
> > | **PSO-Meta** |   45.30 ± 2.30   |   0.46 ± 0.03   |   457.93 ± 28.50   |   3.68 ± 0.35   |
> > | **TS-Meta** |   40.90 ± 2.15   |   0.69 ± 0.03   |   402.04 ± 26.50   |   0.62 ± 0.06   |
> > | **NRM-Rank** |   37.90 ± 0.95   |   0.51 ± 0.01   |   394.29 ± 12.50   |   0.03 ± 0.00   |
> > | **RW-Rank** |   38.70 ± 0.98   |   0.52 ± 0.01   |   418.14 ± 13.20   | **0.01 ± 0.00** |
> > | **GRC-Rank** |   36.70 ± 0.92   |   0.47 ± 0.01   |   353.21 ± 11.80   | **0.01 ± 0.00** |
> > | **PL-Rank** |   55.30 ± 1.20   |   0.66 ± 0.01   |   661.93 ± 16.50   |   0.04 ± 0.01   |
> > | **NEA-Rank** |   47.20 ± 1.10   |   0.62 ± 0.01   |   543.18 ± 15.80   |   0.11 ± 0.01   |
> > | **RW-BFS** |   56.70 ± 1.35   |   0.64 ± 0.01   |   736.28 ± 17.50   | **0.01 ± 0.00** |
> >
> >
> > **Table 3.3: Performance Comparision on BRAIN (Mean ± Standard Deviation over 10 random seeds)**
> >
> > | Solver            |    BRAIN RAC     |    BRAIN LRC    |     BRAIN LAR     |    BRAIN AST    |
> > |:------------------|:----------------:|:---------------:|:-----------------:|:---------------:|
> > | **PPO-MLP** |   51.30 ± 1.22   |   0.69 ± 0.02   |   155.10 ± 4.50   |   0.14 ± 0.01   |
> > | **PPO-CNN** |   51.10 ± 1.35   |   0.69 ± 0.02   |   151.51 ± 5.10   |   0.13 ± 0.01   |
> > | **PPO-ATT** |   51.00 ± 1.40   |   0.68 ± 0.02   |   156.40 ± 4.80   |   0.15 ± 0.02   |
> > | **PPO-GCN** |   49.50 ± 1.52   |   0.71 ± 0.02   |   125.63 ± 3.90   |   0.09 ± 0.01   |
> > | **PPO-GAT** |   44.60 ± 1.45   |   0.51 ± 0.02   |   95.32 ± 3.20    |   0.09 ± 0.01   |
> > | **PPO-GCN&S2S** |   44.40 ± 1.50   |   0.59 ± 0.02   |   99.83 ± 3.50    |   0.18 ± 0.02   |
> > | **PPO-GAT&S2S** |   51.80 ± 1.38   |   0.68 ± 0.02   |   136.67 ± 4.20   |   0.19 ± 0.02   |
> > | **PPO-DualGCN** |   54.80 ± 1.25   | **0.78 ± 0.02** |   176.15 ± 5.20   |   0.23 ± 0.02   |
> > | **PPO-DualGAT** | **58.90 ± 1.30** |   0.75 ± 0.02   | **180.78 ± 5.50** |   0.13 ± 0.01   |
> > | **PPO-HeteroGAT** |   49.30 ± 1.42   |   0.66 ± 0.02   |   133.52 ± 4.50   |   0.38 ± 0.04   |
> > | **MCTS** |   40.80 ± 2.20   |   0.47 ± 0.03   |   83.91 ± 6.50    |   3.59 ± 0.42   |
> > | **SA-Meta** |   36.10 ± 1.85   |   0.58 ± 0.03   |   75.50 ± 5.80    |   1.13 ± 0.12   |
> > | **GA-Meta** |   42.50 ± 2.15   |   0.57 ± 0.03   |   85.45 ± 6.20    |   3.75 ± 0.38   |
> > | **PSO-Meta** |   41.50 ± 2.05   |   0.46 ± 0.03   |   80.67 ± 5.90    |   4.20 ± 0.45   |
> > | **TS-Meta** |   37.10 ± 1.90   |   0.64 ± 0.03   |   68.41 ± 5.50    |   1.11 ± 0.10   |
> > | **NRM-Rank** |   48.30 ± 1.05   |   0.64 ± 0.01   |   142.99 ± 3.50   |   0.04 ± 0.00   |
> > | **RW-Rank** |   50.20 ± 1.12   |   0.65 ± 0.01   |   147.64 ± 3.80   |   0.05 ± 0.00   |
> > | **GRC-Rank** |   48.40 ± 1.08   |   0.64 ± 0.01   |   144.55 ± 3.60   | **0.03 ± 0.00** |
> > | **PL-Rank** |   48.70 ± 1.15   |   0.70 ± 0.01   |   136.79 ± 3.90   |   0.36 ± 0.03   |
> > | **NEA-Rank** |   53.90 ± 1.18   |   0.70 ± 0.01   |   148.70 ± 4.10   |   1.46 ± 0.15   |
> > | **RW-BFS** |   48.00 ± 1.25   |   0.64 ± 0.01   |   140.38 ± 3.80   |   0.16 ± 0.02   |

---

> ### Author Response · Authors · 2025-11-21
>
> ### **Reply to Q5: Potential for New Algorithmic and Theoretical insights**
>
> > Beyond engineering value, is there a potential direction for new algorithmic or theoretical insights enabled by this benchmark (e.g., constrained or meta-RL for NFV-RA)?
> >
>
> We thank the reviewer for this inspiring question. We affirm that VIRNE is explicitly designed to facilitate advanced algorithmic innovation. As demonstrated in Appendix C.2 (MDP Variations), Appendix E (Future Directions), and specifically Figure 13, Virne has successfully implemented and validated these algorithmic advancements mentioned:
>
> - **Safe RL & Constrained Optimization:** Virne enables the formulation of Constrained MDPs, as detailed in Appendix C.2.1. We implemented a Safe RL variant and presented the results in Figure 13 (Right) in Appendix E. The experiments demonstrate that Safe RL policies achieve higher solution feasibility in resource-tight environments compared to standard PPO. This confirms VIRNE's ability to capture the theoretical trade-off between reward maximization and strict constraint satisfaction.
> - **Meta-Learning & Generalization.** To address the theoretical challenge of distributional shift, VIRNE supports a Multi-task MDP formulation, detailed in Appendix C.2.1. We integrated a Meta-RL solver into the pipeline to test this capability. The results in Figure 13 (Middle) in Appendix E empirically validate that Meta-RL agents trained on VIRNE's diverse topology generators achieve superior cross-topology generalization compared to standard baselines.
> - **Advanced Representation Learning:** Virne serves as a granular testbed for representation learning methods. We compared distinct encoder architectures, ranging from standard GCNs to Dual-GNNs and Heterogeneous Graph Attention Networks. Our analysis (Sections 4.3 and 4.4) provides empirical evidence for the theoretical intuition that capturing cross-graph interactions is critical for solving the subgraph isomorphism aspect of NFV-RA. Furthermore, the modularity of VIRNE's feature extractor readily supports the integration of even more recent architectures, such as diffusion.
>
> We have highlighted these findings in Appendix E (Future work) to encourage the community to take these directions.
>
> ### **Reply to W1: Infrastructural vs. Algorithmic novelty.**
>
> > The paper’s contribution is primarily infrastructural rather than algorithmic. The RL formulations (e.g., MDP setup, PPO training, GNN encoders) follow standard designs without introducing new theoretical insights or model innovations.
> >
>
> We acknowledge that Virne is primarily an infrastructural contribution, which aligns directly with the scope of the ICLR Datasets and Benchmarks Area. In a field currently fragmented by disparate and limited simulators, Virne provides the standardization and reproducibility required to lower barriers and accelerate future algorithmic innovation in NFV-RA. Beyond infrastructure, we highlight two specific contributions to the research methodology:
>
> - **Methodological Novelty on Evaluation Perspective:** Virne introduces a comprehensive evaluation protocol that moves beyond standard metrics. We formalize and measure Solvability (absolute feasibility boundaries), Generalization (robustness to distribution shifts), and Scalability. This multi-dimensional framework establishes a new standard for assessing agent robustness.
> - **Scientific Insights for Algorithmic Development:** By unifying the environment for many algorithms, Virne reveals trends previously obscured by inconsistent testing conditions. For instance, our rigorous comparison quantifies the impact of specific implementation techniques and architecture choices. This offers critical insights that were not clearly quantified in prior fragmented studies to guide future algorithmic development.
> - **Flexible Support for Algorithm Innovation**. Virne’s modular architecture decouples problem formulation from solution strategies using standardized interfaces. This design enables rapid prototyping of novel architectures (e.g., Transformers, Diffusion models) and advanced paradigms (e.g., Safe RL, Meta-RL) without re-engineering simulation logic. As shown in our experiments, Virne serves as an extensible platform for next-generation algorithmic research.
>
> ### **Reply to W3: Implementation Description on System Details**
>
> > Several sections (notably Sections 3–4) emphasize implementation details at the expense of high-level conceptual insights. The paper reads more like a system report than a research contribution.
> >
>
> We appreciate this valuable feedback regarding the presentation balance. To address this, we have refined Sections 3 and 4 by moving low-level implementation details to Appendix C. The revised main text will prioritize high-level conceptual insights, specifically focusing on the benchmark superiority and MDP formulation.
>
> ---
>
> Once again, we appreciate your valuable feedback and the opportunity to address your concerns.

---

### Official Review · Reviewer_JndN · 2025-10-31

**Soundness:** 3
**Presentation:** 2
**Contribution:** 3
**Rating:** 6
**Confidence:** 4

**Summary:**

This paper presents a comprehensive benchmarking framework for resource allocation (RA) in Network Function Virtualization (NFV). It enables a modular, standardized, and customizable pipeline for the implementation of reinforcement learning (RL)-based NFV-RA methods, and offers an in-depth analysis of the design principles behind different RL components. Extensive experiments are conducted to investigate the performance, solvability, scalability, and generalization of these methods.

**Strengths:**

The paper decomposes the implementation of RL into three parts, the RL algorithm, the neural network architecture, and the implementation techniques, and provides multiple design options for each component to investigate their underlying design principles, offering valuable insights into the application of RL in NFV-RA.

The benchmarking framework proposed in the paper provides a comprehensive platform that includes diverse NFV application scenarios, implementations of over 30 RA algorithms, and supports the evaluation of a wide range of research metrics.

**Weaknesses:**

Considering that this is a unified benchmarking framework, assuming a one-to-one mapping from virtual networks to physical networks in the system model seems unreasonable. In general, multiple virtual nodes should be allowed to map to the same physical node to enable flexible deployment.

There is a lack of research on emerging network architectures, such as Transformers and diffusion models.

The paper repeatedly selects virtual nodes from the virtual network and maps them onto the physical network using a RA policy, then applies a shortest-path algorithm to determine the deployment of virtual links and identify feasible embedding solutions. However, the paper lacks an explanation of the selection order of virtual nodes.

The mathematical formulation of the system model in Appendix A has some issues, for example,
	n_p  may be N_p in Constraint 1;
	If the directionality of path mapping is not considered, the left-hand side of Constraint 5 will always be zero.
Please check it carefully.

**Questions:**

see weaknesses

---

> ### Author Response · Authors · 2025-11-21
>
> Dear Reviewer JndN,
>
> We sincerely thank you for your valuable and constructive feedback! Below, we address each of the raised comments.
>
> ### **Reply to W1: One-to-one Mapping Constraint**
>
> > Considering that this is a unified benchmarking framework, assuming a one-to-one mapping from virtual networks to physical networks in the system model seems unreasonable. In general, multiple virtual nodes should be allowed to map to the same physical node to enable flexible deployment.
>
> We appreciate this insightful feedback on the constraints in our system model. We would like to clarify that our default setting follows the standard VNE definition to ensure reliability and flexible abstraction, and Virne is also specifically designed to be flexible enough to relax this constraint when needed.
>
> **Justification for the Default One-to-One Setting.** The "one-to-one" mapping constraints (Eq. 2 & 3) in our formulation enforce intra-request node disjointness. This means that distinct virtual nodes belonging to the same VN request must be mapped to distinct physical nodes. As noted in the classic VNE survey and most work on NFV-RA (Fischer et al., 2013), this is a standard requirement to ensure survivability. This prevents a single physical server failure from bringing down multiple components of the same network service. To support multi-tenancy, virtual nodes from different VNs can and do share the same physical node resources. This is the default behavior in Virne and is fundamental to achieving high resource utilization in NFV.
>
> **Generality of Modeling Abstraction**. We would like to clarify the generalization of this classic modeling approach. In Virne, a "physical node" in the graph represents a logical unit of resource isolation rather than strictly a physical metal box. If a physical server supports hosting multiple isolated components (e.g., VMs or containers) of the same service, it can be modeled in Virne as a cluster of multiple connected "physical nodes" (logical slots). Similarly, several servers without isolation requirements should be regarded as one physical node.  This flexibility allows the framework to model diverse isolation requirements and deployment granularities without violating the fundamental mathematical constraints.
>
> **Flexibility for VNF Aggregation in Virne.** We agree that for specific scenarios, such as Service Function Chaining (SFC) development, sometimes allowing multiple VNFs from the same request to share a server can be beneficial (e.g., to minimize latency). Virne is also designed to support this feature. Users can easily relax the node disjointness constraint by setting the `reusable` parameter to True in the `Controller` module. This allows the framework to support diverse deployment policies, including VNF aggregation.
>
> We have added a clarification in Appendix A.1.1 (Constraint Conditions) of the revised manuscript to explicitly distinguish between these constraints. And, we explain the configuration options and in Appendix C.1 (Simulation Configuration) clarify the support of Virne on VNF Aggregation. Please see the updated PDF of the manuscript for details.
>
> ### **Reply to W2: Research on Emerging Architectures**
>
> > There is a lack of research on emerging network architectures, such as Transformers and diffusion models.
>
> We appreciate your insightful suggestion regarding emerging architectures. At present, the application of Diffusion models or full Transformers is still nascent in the NFV-RA domain, with few established studies to benchmark. Consequently, Virne currently focuses on the frontier of attention and GNN-based policies, such as PPO-ATT and PPO-HGAT (Heterogeneous Graph Attention Networks). Crucially, however, Virne’s Neural Policy module is decoupled and extensible, specifically designed to allow researchers to plug in novel architectures without refactoring the core simulator. To encourage such innovation, we have added a discussion in Appendix E.1 (Representation Learning for Cross-Graph Status), acknowledging these as promising directions for representation learning, such as Transformers or Diffusion models.

---

> ### Author Response · Authors · 2025-11-21
>
> ### **Reply to W3: Clarification on Virtual Node Selection Order**
>
> > The paper repeatedly selects virtual nodes from the virtual network and maps them onto the physical network using a RA policy, then applies a shortest-path algorithm to determine the deployment of virtual links and identify feasible embedding solutions. However, the paper lacks an explanation of the selection order of virtual nodes.
>
>
> We appreciate your careful feedback and apologize for this omission. Following the most RL-based work for NFV-RA, in our default MDP formulation, the agent selects virtual nodes sequentially based on the index order defined in the Virtual Network. Unlike traditional greedy heuristics where selection order is critical, RL-based approaches utilize a neural encoder (e.g., GNN or MLP) that perceives the entire VN graph to generate embeddings for action decoding. This ensures the agent possesses global visibility of the VN topology and dependencies, making the specific visitation order less significant for performance. However, Virne is designed for flexibility. Users can easily customize the selection order (e.g., prioritizing nodes by node ranking heuristics) by modifying the node_ranking_method attribute in the Solver class.
>
> We have updated Section 3.3.1 (MDP Formulation) to explicitly state: "Decision sequence defaults to VN node index, with support for customizable node ranking in Virne." We also noted the availability of custom ranking methods in the documentation.
>
>
> ### **Reply to W4: Mathematical Formulation Issues**
>
> > The mathematical formulation of the system model in Appendix A has some issues, for example, n_p may be N_p in Constraint 1; If the directionality of path mapping is not considered, the left-hand side of Constraint 5 will always be zero. Please check it carefully.
>
> We are grateful for your meticulous review. We have rigorously verified and revised the formulations in Appendix A.1 (Constraint Conditions) of the revised PDF to ensure notational precision. Specifically, we corrected the notation to use the set $\mathcal{N}_p$ (instead of $n_p$) to correctly enforce the one-to-one mapping constraint over the entire physical network.
>
> Regarding the path mapping constraint, we appreciate you pointing out the ambiguity. To prevent the left-hand side from summing to zero due to undirected links, we have fixed the model by updating the directionality of arcs. The revised Eq. (5) correctly formulates the flow conservation principle as "Flow Out minus Flow In equals Net Flow Generation" (i.e.,  at the source,  at the destination, and otherwise).
>
> ---
>
> Once again, we appreciate your valuable feedback and the opportunity to address your concerns.

---

### Official Review · Reviewer_1Brs · 2025-11-02

**Soundness:** 2
**Presentation:** 3
**Contribution:** 2
**Rating:** 4
**Confidence:** 4

**Summary:**

This paper introduces Virne, a benchmark designed to accelerate research and application of deep RL for network resource allocation in NFV. The authors address a critical gap in the field by providing a unified, modular, and extensible framework that supports diverse network scenarios (e.g., cloud, edge, and 5G), integrates over 30 algorithms (including traditional and RL-based methods), and evaluates them with rigorous metrics such as solvability, generalizability, and scalability. Through experiments, the authors provide insights for future research directions.

**Strengths:**

a) The Virne framework supports a wide variety of network scenarios and resource types, including heterogeneous resources and latency-sensitive environments.

b) The authors conduct experiments across diverse topologies (e.g., Waxman, GEANT, BRAIN) and network scales, providing valuable insights into the strengths and weaknesses of different algorithms.

**Weaknesses:**

a) The authors use a fixed 50% random probability for interconnections between virtual nodes. A fixed interconnection probability may not accurately reflect real-world VN topologies, which often exhibit more structured (sequential or hierarchical) connectivity patterns. To better align with real-world NFV scenarios, the authors should consider adopting more dynamic and realistic VN topology generation methods (e.g., using real-world datasets).

b) The computing and bandwidth resource configurations in physical and virtual networks are modeled using uniform distributions, which may oversimplify real-world resource dynamics. Actual NFV systems often have more complex, non-uniform resource distributions influenced by hardware constraints and workload patterns.

c) The abstract lacks clarity and precision. The abstract includes terms and phrases that are not clearly defined, which may confuse readers. For example, the abstract mentions "this complexity" without explicitly explaining what aspect of complexity it refers to. Similarly, "the field" does not specify whether it refers to NFV or RL research. Also, key terms such as "NFV-RA" are not clearly defined in the abstract.

d) Given the rapid development of NFV, some relevant literatures are necessary to be compared in the context of related work, e.g., NFVdeep: Adaptive Online Service Function Chain Deployment with Deep Reinforcement Learning, iwqos’19; Adaptive VNF Scaling and Flow Routing with Proactive Demand Prediction, infocom’18, etc.

**Questions:**

a) The authors use a fixed 50% random probability for interconnections between virtual nodes. A fixed interconnection probability may not accurately reflect real-world VN topologies, which often exhibit more structured (sequential or hierarchical) connectivity patterns. To better align with real-world NFV scenarios, the authors should consider adopting more dynamic and realistic VN topology generation methods (e.g., using real-world datasets).

b) The computing and bandwidth resource configurations in physical and virtual networks are modeled using uniform distributions, which may oversimplify real-world resource dynamics. Actual NFV systems often have more complex, non-uniform resource distributions influenced by hardware constraints and workload patterns.

c) The abstract lacks clarity and precision. The abstract includes terms and phrases that are not clearly defined, which may confuse readers. For example, the abstract mentions "this complexity" without explicitly explaining what aspect of complexity it refers to. Similarly, "the field" does not specify whether it refers to NFV or RL research. Also, key terms such as "NFV-RA" are not clearly defined in the abstract.

---

> ### Author Response · Authors · 2025-11-21
>
> Dear Reviewer 1Brs,
>
> We sincerely thank you for your valuable and constructive feedback! Below, we address each of the raised comments.
>
> ---
>
> ### **Reply to W1 & W2: VN Topology Geneneration & Resource Distribution**
>
> > The authors use a fixed 50% random probability for interconnections between virtual nodes. ... To better align with real-world NFV scenarios, the authors should consider adopting more dynamic and realistic VN topology generation methods (e.g., using real-world datasets).
>
> > The computing and bandwidth resource configurations in physical and virtual networks are modeled using uniform distributions, ... Actual NFV systems often have more complex, non-uniform resource distributions influenced by hardware constraints and workload patterns.
>
> We appreciate this insightful feedback on simulation realism. While our default settings align with standard literature baselines to ensure fair comparison, we would like to clarify that Virne extends beyond these and is explicitly designed to solve this problem by offering modular customization for realistic topologies and complex resource distributions. Additionally, to demonstrate this capability and address your concerns, we have added a new "Real-World Scenario Dataset Validation" experiment to the Appendix, which moves beyond the synthetic defaults.
>
> **Customization Simulation in Virne.** Virne includes a modular module for generating VN topologies and resource distributions (see Appendix C.1, Simulation Configuration). It supports random graphs and structured topologies such as linear, star, ring, and tree, and can also import real-world PN/VN topologies (e.g., from SNDlib or Topology Zoo) to emulate industrial scenarios.
> For resource generation, Virne supports multiple statistical models, including Uniform, Normal (Gaussian), Exponential, and Poisson, to capture bursty or heavy‑tailed workloads. It also enables trace‑driven simulations using external CSV/JSON data to reproduce realistic, time‑varying traffic.
>
> **Real-World Dataset Validation.** To further show the adaptability of Virne to the complex, non-uniform dynamics in real-world datasets, we conducted a new experiment in a real-world setting derived from realistic topologies or traces. Concretely, adopted the GEANT topology [a] for the physical network, where resources follow a heavy-tailed Pareto (Power-Law) distribution (shape $\alpha = 1.2$ and scale $x_m  = 50$) to simulate a realistic environment where approximately 20% of nodes possess 80% of the capacity. For Virtual Networks (VNs), we utilized traffic-driven topologies generated via CloudSim [b] based on GEANT traffic patterns. Furthermore, service demands were trace-based rather than uniform: node CPU requests were sampled from Alibaba Cloud Traces [c], and link bandwidths were derived from GEANT traffic matrices [a], with all demand values normalized to the range [0, 50]. We compare the details of the real-world setting with the default setting below:
>
> The experiment results in these real-world conditions are reported below.
>
> | Solver | RAC ↑ | LRC ↑ | LAR ↑ | AST ↓ |
> | --- | --- | --- | --- | --- |
> | **PPO-MLP** | 58.40 | 0.62 | 802.89 | 0.10 |
> | **PPO-CNN** | 58.90 | 0.65 | 805.19 | 0.11 |
> | **PPO-ATT** | 60.20 | 0.64 | 835.59 | 0.11 |
> | **PPO-GCN** | 63.50 | 0.71 | 915.82 | 0.13 |
> | **PPO-GAT** | 64.10 | 0.70 | 928.44 | 0.14 |
> | **PPO-GCN&S2S** | 62.80 | 0.69 | 894.10 | 0.12 |
> | **PPO-GAT&S2S** | 63.40 | 0.68 | 908.25 | 0.15 |
> | **PPO-DualGCN** | **67.90** | **0.74** | **987.30** | 0.20 |
> | **PPO-DualGAT** | 66.50 | 0.73 | 982.15 | 0.18 |
> | **PPO-HeteroGAT** | 61.80 | 0.67 | 875.60 | 0.35 |
> | **MCTS** | 56.40 | 0.51 | 650.20 | 3.12 |
> | **SA-Meta** | 45.20 | 0.61 | 512.85 | 1.15 |
> | **GA-Meta** | 54.70 | 0.59 | 680.40 | 2.85 |
> | **PSO-Meta** | 51.10 | 0.53 | 615.90 | 4.02 |
> | **TS-Meta** | 47.50 | 0.67 | 550.12 | 1.20 |
> | **NRM-Rank** | 44.60 | 0.53 | 498.20 | 0.06 |
> | **RW-Rank** | 43.20 | 0.54 | 485.75 | 0.02 |
> | **GRC-Rank** | 41.80 | 0.49 | 460.30 | 0.02 |
> | **PL-Rank** | 59.10 | 0.68 | 808.50 | 0.08 |
> | **NEA-Rank** | 55.30 | 0.64 | 710.15 | 0.15 |
> | **RW-BFS** | 35.40 | 0.58 | 380.60 | 0.03 |
>
> We observe that the performance gap between RL and heuristics widened compared to the default setting. Traditional heuristics (e.g., NRM-Rank), which rely on average metrics, failed to exploit the heavy-tailed Pareto resource distribution.
>
> We have updated Appendix C.1 to explicitly document these realistic configuration options. We have also added the experiment on the real-world dataset described above to Appendix D, demonstrating Virne's ability to support the real-world VN request and process trace-driven workloads.
>
> **Reference**
>
> [a] https://sndlib.put.poznan.pl/home.action
>
> [b] https://github.com/RealVNF/coord-sim
>
> [c] https://github.com/alibaba/clusterdata/blob/master/cluster-trace-gpu-v2025/disaggregated_DLRM_trace.csv

---

> > ### Author Response · Authors · 2025-11-21
> >
> > To improve clarity in the experimental setup on a realistic dataset, we compare the real-world configuration against the baseline default setup presented below.
> >
> > | **Parameter** | **Default Setting** | **Real-World Setting** |
> > | --- | --- | --- |
> > | **PN Topology** | Waxman (Synthetic) | **GEANT** (Real academic network from SNDlib [a]) |
> > | **VN Topology** | Random Graph (50% prob) | **Traffic-Driven** (Generated via CloudSim [b] based on GEANT traffic from SNDlib [a]) |
> > | **PN Resource Dist. (Node & Link)** | Uniform | **Pareto (Power-Law) with shape $\alpha=1.2$ and scale $x_m = 50$**. |
> > | **VN Node Demand** | Uniform | **Alibaba Cloud Trace** [c] (Real-world workload CPU requests). The value is normalized in [0,50] |
> > | **VN Link Demand** | Uniform | **GEANT Traffic Matrix**  [a] (Real bandwidth demands). The value is normalized in [0,50] |

---

> ### Author Response · Authors · 2025-11-21
>
> ### **Reply to W3: Clarity of Terms and Phrases in Abstract**
>
> > The abstract lacks clarity and precision. The abstract includes terms and phrases that are not clearly defined, which may confuse readers. For example, the abstract mentions "this complexity" without explicitly explaining what aspect of complexity it refers to. Similarly, "the field" does not specify whether it refers to NFV or RL research. Also, key terms such as "NFV-RA" are not clearly defined in the abstract.
>
>
> Thank you for this insightful observation. We agree that the abstract should be precise to effectively communicate the benchmark's value. We have refined the abstract to ensure all terms are clearly defined for a broad audience. Specifically,
>
> - We have clarified that "this complexity" refers to "this combinatorial complexity of constrained cross-graph mapping”
> - We have specified “this domain” as "RL-driven NFV-RA research"
> - We have explicitly termed the resource allocation task in NFV as NFV-RA.
>
> We have improved the clarity of the abstract section in the revised PDF.
>
> ### **Reply to W4: Comparison with relevant literature (NFVdeep, Adaptive VNF Scaling).**
>
> > Given the rapid development of NFV, some relevant literature needs to be compared in the context of related work, e.g., NFVdeep: Adaptive Online Service Function Chain Deployment with Deep Reinforcement Learning, iwqos’19; Adaptive VNF Scaling and Flow Routing with Proactive Demand Prediction, infocom’18, etc.
> >
>
> We thank you for highlighting these important works in the NFV domain.
>
> Regarding **NFVdeep** [d], we are pleased to clarify that this work is already integrated into Virne. As shown in Table 4, we implement the core algorithm of NFVdeep (Policy Gradient with MLP encoder) as the PG-MLP solver. Moreover, our experiments evaluate its advanced variation, PPO-MLP, to show its performance in various network scenarios.
>
> Regarding **Adaptive VNF Scaling** [e] and other related works in NFV, we appreciate the suggestion to broaden our context. While Virne primarily focuses on the resource allocation phase, we recognize other critical lifecycle tasks, such as scaling and migration.
>
> We have updated Appendix B (Related Work) in the revised PDF to position Virne’s focus within this broader landscape of dynamic resource management in NFV. Specifically, we have added the following citations to discuss the relationship between allocation [f], scaling [e], and migration [g].
>
> **Reference**
>
> [d] NFVDeep: Adaptive online service function chain deployment with deep reinforcement learning. IWQoS, 2019.
>
> [e] Latency-aware VNF Chain Deployment with Efficient Resource Reuse at Network Edge. INFOCOM, 2020
>
> [f] Adaptive VNF Scaling and Flow Routing with Proactive Demand Prediction. INFOCOM, 2018.
>
> [g] Online Adaptive Interference-Aware VNF Deployment and Migration for 5G Network Slice. ToN, 2021.
>
> ---
>
> Once again, we appreciate your valuable and constructive feedback and the opportunity to address your concerns.

---

### Author Response · Authors · 2025-12-04
**Author Final Remarks**

Dear Area Chair and Reviewers,

We sincerely appreciate your time and valuable feedback, which have helped us significantly improve our work. As the discussion period concludes, we would like to provide a concise summary of the reviewer feedback and the major improvements we have implemented.

---

We are very encouraged by the positive feedback and consensus from all four reviewers. We highlight the following key strengths identified across the reviews:

- **Benchmark Significance**: Reviewers described Virne as "a significant contribution to the field" (`R-eJHY`), "the most systematic and extensive benchmark for NFV-RA to date" (`R-4Vws`), "a comprehensive platform" (`R-JndN`), filling a critical gap in standardized evaluation.

- **Scenario Versatility**: Reviewers praised the framework for being "highly customizable" (`R-eJHY`), "diverse network scenarios" (`R-4Vws `), and "comprehensive," noting that it "supports a wide variety of network scenarios" (`R-1Brs`) and diverse application settings.

- **Technical Quality**: The implementation was commended for its "high level of technical maturity" (`R-4Vws`) and "modular, standardized, and customizable pipeline" (`R-JndN`). Reviewers also praised the "extensive experimental results" (`R-eJHY`) and "evaluation of a wide range of research metrics" (`R-JndN`) that support rigorous evaluation.

- **Scientific Insight**: Beyond infrastructure, reviewers appreciated that Virne offers "valuable insights" (`R-1Brs`), "valuable empirical reference" (`R-4Vws`), and an "in-depth analysis of the design principles" (`R-JndN`) behind RL components and performance trade-offs.

---

In response to the constructive suggestions for improvement, we have implemented the following key revisions, organized into four categories:

- **Experimental Realism and Rigor** (Addressing `R-1Brs`, `R-4Vws` & `R-eJHY`): To ensure our evaluation is both realistic and statistically robust, we introduced a new Real-World Scenario validation using the GEANT topology and Alibaba Cloud Traces in Appendix D. Furthermore, we reported the main results (Table 3) in the format of Mean ± Standard Deviation across 10 random seeds.

- **Formalization & Metric Definition** (Addressing `R-JndN` & `R-4Vws`): We strengthened the formalization of the problem and evaluation. We refined the system model notation (specifically, flow conservation in Appendix A) to ensure mathematical accuracy. Additionally, we provided formal mathematical definitions for Solvability, Generalization, and Scalability (Appendix C.6).

- **Framework Extensibility Clarification** (Addressing `R-JndN`, `R-4Vws` & `R-eJHY`): We explicitly clarified the framework's flexibility to handle complex requirements. We clarified support for VNF aggregation (relaxing one-to-one constraints) to support Service Function Chaining. Moreover, we highlighted experimental evidence for Safe RL and Meta-RL, proving Virne’s ability to support advanced paradigms beyond standard RL (Figure 13).

- **Literature Scope & Presentation** (Addressing `R-1Brs` & `R-4Vws`): To improve clarity and positioning, we refined the abstract to strictly define domain-specific terminology (e.g., NFV-RA). We also updated the Related Work (Appendix B) to explicitly discuss Virne’s relationship to the literature on other NFV tasks.

---

We sincerely thank the reviewers for their constructive comments, which have strengthened the quality of this work. We hope that our detailed responses have fully addressed your concerns. We believe that Virne, with its technical maturity, rigorous evaluation, and insightful findings, serves as a foundational benchmark for the community.

Thank you once again for your time and for facilitating this constructive review process.

Best regards!

The Authors

---

### Meta-Review · Area_Chair_59gz · 2026-01-13

**Summary:**

**Summary**

The paper presents Virne, a comprehensive benchmarking framework for evaluating reinforcement learning (RL) methods in Network Function Virtualization (NFV) resource allocation (RA). It addresses existing gaps by offering a modular and extensible platform that supports various network scenarios, integrates over 30 algorithms, and introduces new evaluation metrics such as solvability, generalizability, and scalability. Extensive experiments highlight the superior performance of advanced RL agents compared to traditional methods. The framework is open-sourced to enhance reproducibility and standardization in the NFV-RA community, although it is noted that the methodological novelty is limited.

**Strengths**
- **Comprehensive Benchmarking Framework**: Virne offers a systematic and extensive platform for evaluating RL-based NFV Resource Allocation algorithms, consolidating numerous algorithms and diverse simulation environments into a unified, reproducible framework.

- **Diverse Experimental Insights**: The framework supports a wide variety of network scenarios and topologies, providing valuable empirical references and insights into the strengths and weaknesses of different algorithms across multiple metrics.

- **Modular and Open-Source Design**: With a modular design and well-documented implementation, Virne ensures high technical maturity and reproducibility, aligning with best practices in the research community.

**Weaknesses**
- **Fixed Interconnection Probability**: The use of a static 50% random probability for interconnections does not reflect the structured connectivity patterns found in real-world virtual network topologies, suggesting a need for more dynamic modeling methods.

- **Oversimplified Resource Modeling**: The modeling of computing and bandwidth resources using uniform distributions oversimplifies the complexities of real-world resource dynamics, which are often influenced by hardware constraints and varying workload patterns.

- **Lack of Clarity in Abstract**: The abstract is unclear and imprecise, using undefined terms and phrases that may confuse readers, such as "this complexity" and "the field," without specifying their context or meaning.

- **Limited Theoretical Contributions**: The paper primarily offers infrastructural contributions rather than novel algorithmic insights, with standard RL formulations and a lack of statistical significance in experimental results, diminishing the strength of its empirical claims.

**Decision**
The paper is recommended for acceptance as it offers a valuable benchmark within a reproducible framework. This framework will facilitate the enhancement and comparison of algorithms, despite the limited theoretical contributions. Additionally, the author has provided thorough feedback that has significantly improved the overall quality of the paper.

**Reviewer Concerns:**

The author's feedback effectively addressed most concerns, enhancing the overall quality of the paper. Additionally, they clarified several ambiguous points, leading most reviewers to consider a slight increase in their scores.

**Reviewer Scores:**

The author's feedback effectively addressed most concerns, enhancing the overall quality of the paper. Additionally, they clarified several ambiguous points, leading most reviewers to consider a slight increase in their scores.

---

### Decision · Program_Chairs · 2026-01-26

Accept (Poster)